

# Sediment aggradation rates for Himalayan Rivers revealed through SAR remote sensing

Jingqiu Huang[1], Hugh D. Sinclair[1]

[1]School of GeoSciences, University of Edinburgh, Edinburgh, EH8 9XP, UK

*Correspondence to*: Jingqiu Huang (jq.huang@ed.ac.uk)

**Abstract.** This study uses Synthetic Aperture Radar (SAR) to quantify sediment aggradation rates in the proximal gravel-rich portions of the rivers that drain out of the Himalayan Mountain Front onto the Gangetic Plains. Implementing the Small Baseline Subset (SBAS) InSAR (Interferometric SAR) method on Sentinel-1 C-band InSAR residual topographic phase, we measure millimeter-scale elevation changes during the period from 2016 to 2021 covering ~15 km reaches of four rivers from

the mountain front downstream to the gravel-sand transition. This is the first study to apply differential residual topographic phase mapping seasonally dry (ephemeral) rivers. Results indicate sediment aggradation in river channels that accumulates during the wet monsoon, with rates reaching up to approximately 20 mm/yr (i.e., per monsoon) near the mountain front, decreasing to nearer zero downstream of the gravel-sand transition. Meanwhile, the floodplain in the basin is subsiding at varying rates that average ~15 mm/yr. These findings enable a temporal understanding of sediment aggradation rates that

impact river avulsion and flood risk, particularly for the rapidly growing rural communities in Nepal and Bihar, India. Our study demonstrates the feasibility of InSAR techniques in geomorphological monitoring that can act as input into flood risk modelling and management in the Gangetic Plains.

## 1 Introduction

As mountain rivers discharge into surrounding plains, they build large accumulations of coarse sediment to form alluvial fans

or cones with a convex cross-sectional topography (Bull, 1977). The sediment that accumulates at the mountain front is associated with the transition from river channels that flow over tectonically uplifting bedrock in the range to subsiding foreland basins (Flemings and Jordan, 1989; Sinclair et al., 1991); this transition is associated with reductions in channel gradients and abrupt deposition of coarse sediment loads. Sediment deposition at mountain fronts is usually characterised by coarse gravel-rich deposition until, at some distance downstream there is an abrupt reduction in grain size at the 'gravel-sand

transition' (Dingle et al., 2021; Sambrook Smith and Ferguson, 1995). The position of the gravel to sand transition is interpreted to be limited by the total flux of coarse bedload discharged to the Gangetic Plains (Dingle et al., 2017).

The region of the Himalayan Mountain Front is critically sensitive to sediment supplied from the mountain range as it impacts hydropower and irrigation infrastructure, and modifies flood risk downstream for some of the most vulnerable communities (Dingle et al., 2020; Sinha et al., 2005). However, sediment yield from the range is poorly constrained on a decadal scale, and





known changes in climate and land-use including deforestation, changing agriculture and urbanisation are likely to be impacting these rates (Asselman et al., 2003; Sam and Khoi, 2022). Extreme discharge events, linked to glacial melting, landslide damming and cloudburst events are capable of transporting huge volumes of sediment downstream within the mountains (Graf et al., 2024; Shugar et al., 2021) and then exporting this sediment over the Gangetic Plains (Quick et al., 2023). Hence, being able to monitor changes in sediment budgets is a top priority for these regions.

The rapid accumulation of coarse sediment in the proximal parts of the Himalayan foreland basin have resulted in a long history of channel avulsion (Chakraborty et al., 2010; Sinha et al., 2005), where rivers channels aggrade at a faster rate than their surrounding floodplains leading to super-elevation. Typically, super-elevation occurs when the riverbed approximates the height of the surrounding floodplain and the topographic slope perpendicular to the channel is greater than the long channel gradient (Jerolmack and Mohrig, 2007; Slingerland and Smith, 2004). Channel avulsion results in major flooding and

displacement of rural communities in the Gangetic Plains. The Kosi floods of 2008 were caused by a breakout of the embanked Kosi River across its large fan system displacing ~2.5 million people (Sinha, 2009). An underlying cause of this event was sediment accumulation in the embanked channel elevating it above the floodplains of Bihar State (Mishra and Sinha, 2020). Understanding where channels are prone to avulsion requires knowledge of sediment aggradation rates within channels relative to surrounding floodplains. However, measurements of channel aggradation rates on 10 to 100 year timescales are challenging.

Multi-temporal digital elevation models have proved valuable for quantifying geomorphic change and sediment budgets in some river systems (Wheaton et al., 2010; Williams, 2012). Similarly, repeat surveys of channel bathymetry can demonstrate longer term changes due to erosion and sedimentation (Lane et al., 1994). Photogrammetry used to construct topography based on structure from motion can correct for water depths to enable river bathymetry to be approximated (Shintani and Fonstad, 2017). However, these approaches are not appropriate for the scale of large Himalayan Rivers.

Remote sensing techniques are increasingly capable of recording hydraulic and geomorphic change over large areas in river systems at high temporal resolutions (Rossi et al., 2023). A number of studies have used synthetic aperture radar (SAR) to monitor changes in channel morphology and characterise morphological characteristics such as grainsize (Lin et al., 2023; Olen and Bookhagen, 2020; Purinton and Bookhagen, 2020). In this study, we apply Interferometric Synthetic Aperture Radar (InSAR) to study relative elevation changes of ephemeral gravel riverbeds and their surrounding floodplains as they flow

southward from the Himalayan mountain front in south-eastern Nepal (Fig. 1). The methodology uses the Small Baseline Subset (SBAS) InSAR method on Sentinel-1 C-band SAR data from the ESA (European Space Agency); this approach uses a stack of InSAR phase images to get the millimetre scale elevation change values. In addition to sediment accumulation in river channels, the probable controls on changes in surface elevation in this region are tectonic processes due to thrust propagation (Lavé and Avouac, 2001), regional flexural subsidence (Sinclair and Naylor, 2012) and sediment compaction linked to water

extraction (Huang et al., 2024). In order to isolate the elevation change linked to channel aggradation, we also characterise the background subsidence rates of the floodplains. The results provide the first quantification of sediment aggradation rates across entire channel belts over a distance of ~15 km from the mountain front downstream to the gravel-sand transition (GST).





## 2 Study area

The study area is in Madhesh Province in the southeast of Nepal where we analysed four rivers that drain the Siwalik Hills and are named here rivers 1-4 (Fig. 1). These rivers have relatively small catchments ($20 - 30$ km$^2$) but are thought to have high erosion rates and hence sediment yields as indicated along strike on the Siwalik Hills using detrital cosmogenic nuclides (Mandal et al., 2023). The rivers were selected based on the fact that they are typically dry during the winter months, which ensures that the riverbeds remain undisturbed and suitable for high resolution SAR measurements. The channel widths are

approximately 300 m as they flow across the Plains. The channel slope predominantly aligns in the north-south direction with a gradient of about 0.0003 m/m, which are unlikely to cause geometry distortions in the Sentinel-1 SAR images (Woodhouse, 2017). The analyses of these rivers are dominated by the gravel reach and the downstream gravel to sand transition which has been mapped in this area by Dubille and Lavé (2015). The typical D50 grain size in the channels upstream of the gravel to sand transition is around 2 cm (Dubille and Lavé, 2015; Quick et al., 2020); this is important in terms of determining the

backscatter signal for the InSAR methodology (see Section 3).





**Figure 1: (a) Digital topography of Nepal with study area located. (b) The study area focuses on four rivers (labelled 1 to 4), approximately 76% of the surrounding land is utilized for crop growing. The rivers 1-4 within the InSAR frame (20 km² dark blue box) are approximately 15 km in length, 300 m in width. Rivers and their catchments generated by LSDtopo Tools. They were created based on the easiest flow routes along the lowest point in the channel, with a 30 m sampling rate for the river channel flow**



distance (Mudd et al., 2014), based on a 30 m resolution DEM from http://opentopography.org. Basemap data sources: ESRI World Topographic basemap with hillshade illumination.

# 3 Methodological context

## 3.1 SAR polarimetric backscatter amplitude analysis for dry gravel riverbeds

The backscatter energy (amplitude) is a critical parameter in SAR data analysis as it provides information about the surface scattering properties and is used to derive surface characteristics. Usually, higher backscatter energy results in clearer and more consistent SAR interferograms (Cloude and Papathanassiou, 1998). There are three main backscatter mechanisms: rough surface that causes diffuse scattering, vegetation that causes volume scattering and buildings that cause double-bounce. SAR systems can transmit/receive horizontally (H) or vertically (V) polarized waves. There are four different polarizations in

SAR backscatter signals (HH, HV, VH, VV), which are horizontal transmit and receive (HH), horizontal transmit and vertical receive (HV), vertical transmit and horizontal receive (VH), and vertical transmit and receive (VV), respectively. This configuration is relevant because different polarizations can help classify between diffuse scattering (scattering from rough surfaces), double-bounce scattering (from structures like buildings where the signal bounces twice), and volume scattering (from vegetation or other complex structures), providing applications in forestry, agriculture, and urban mapping (Woodhouse,

2017). For instance, a smooth surface like water might strongly reflect horizontally polarized waves (HH), while rough surfaces might scatter vertically polarized waves (VV) more effectively (Flores-Anderson et al., 2019). The HV and VH polarizations are often used to detect biomass and forest structure since these polarizations are sensitive to volume scattering from vegetation (Mitchard et al., 2009). It is worth noting that Sentinel-1 is a dual-polarized SAR satellite, which means the radar can send and receive both vertical and horizontal waves. However, the VH and HV polarized signals use the same receiving channel;

therefore, VH and HV data are identical (Flores-Anderson et al., 2019).

For diffuse scattering,

$$dB_{VV} > dB_{HH} > dB_{HV} \ , \tag{1}$$

for double bounce scattering,

$$dB_{HH} > dB_{VV} > dB_{HV} \ , \tag{2}$$

and for volume scattering,

$$dB_{HV} > dB_{HH} \ (or \ dB_{VV}) \ , \tag{3}$$

Due to the pebbles in riverbeds, diffuse scattering is stronger in VV polarization, while vegetated gravel bars results in stronger volume scattering in VH polarization. Figure 2 shows that when crossing the dry gravel riverbed on river 2, in the west gravel beds exhibit relatively stronger backscatter amplitude along the VV amplitude transect compared to the east side vegetated



gravel bars. In contrast, along the VH amplitude transect, in the east vegetated gravel bars show relative higher amplitude than the western gravel beds.

SAR raw data is an image, also called Single Look Complex (SLC), which preserve the phase and amplitude information from SAR backscattering signal. Each pixel in the image represents a complex number in Eq. (4). A complex number consists of a real part and an imaginary part, which can be thought of as a vector (a, b) in a two-dimensional plane, with the real part

corresponding to the x-component and the imaginary part to the y-component. It is worth pointing out that the original SLC image is not referenced to a geographic coordinate, but is in azimuth and range project system, which is based on the SAR acquisition geometry. Azimuth is along satellite fly track direction, range is across satellite fly track direction (Loew and Mauser, 2007).

$$S = a + bi \,, \tag{4}$$

Where S is a pixel in an SLC SAR image, a is the real part, and b is the imaginary part.

For amplitude calculated following Eq. (5-8):

$$intensity = a^2 + b^2 \,, \tag{5}$$

$$\sigma = \sqrt{a^2 + b^2} \,, \tag{6}$$

$$\sigma^0 = \sigma * cos\,(\theta) \,, \tag{7}$$

$$dB = 10 * \log_{10}(\sigma^0) \,, \tag{8}$$

Where Θ is the Sentinel-1 satellite SAR acquisition average incident angle. It is worth noting that Eq. (7) is simplified only for incident angle calibration. The returned SAR backscatter energy intensity is usually very small because the radar signal loses strength as it travels to the target and back. The decibel (dB) scale is logarithmic, and the logarithm of a small number (less than 1) is negative. The SAR amplitude values in decibels typically range from about -25 dB to 0 dB for most cases

(Flores-Anderson et al., 2019). Smooth Surfaces (like calm water bodies) tend to have low backscatter and thus lower dB values, often in the range of -25 dB to -20 dB. Vegetated areas typically show moderate backscatter, leading to dB values in the range of -20 dB to -10 dB. Urban areas with many man-made structures, generally have higher backscatter, leading to values from -15 dB to 0 dB or even higher in some cases.

In our study area (Fig. 2), the typical D50 grain size of river sediment upstream of the gravel to sand transition is around 2 cm

(Dubille and Lavé, 2015; Quick et al., 2020). SAR wavelengths that are commonly utilized include the L-band (24 cm), C-band (6 cm), and X-band (3 cm). The C-band and X-band are capable of receiving backscatter from coarse gravels, and cobbles with D50 around 2 cm. In this study, the C-band is chosen due to its free access from the Sentinel-1 SAR satellite. Surface roughness is a relative term, for C-band SAR with 6 cm wavelength, if the cobble's diameter is bigger than 3 cm, which is half of the SAR wavelength, and the surface is considered rough and has a strong SAR backscattering energy (Flores-Anderson et



al., 2019). The median pebble size of 2 cm is considered an intermediate rough surface for C-band SAR, resulting in moderate backscatter intensity.

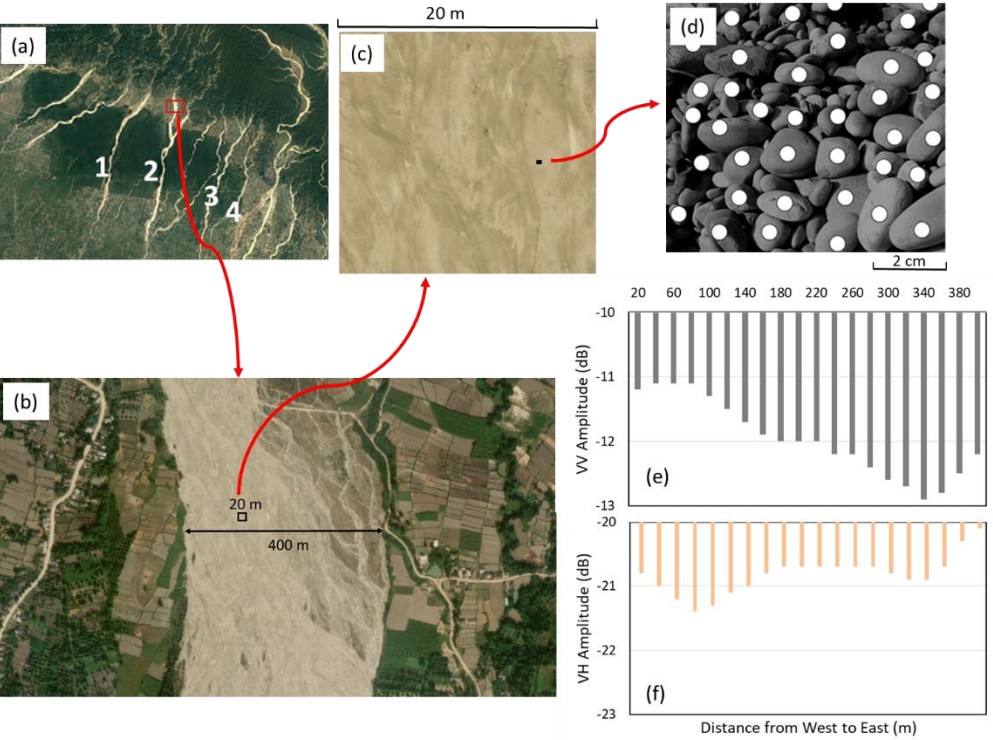

**Figure 2: Illustration of the SAR diffuse backscatter from a dry gravel riverbeds. This illustration demonstrates that each 20 m² InSAR mapping pixel contains tens of thousands of distributed scatters. The intensity of the SAR signal represents the sum of the distributed scatters (white dots in (d)) in that pixel. Some areas may be covered by sand bars, which exhibit lower backscatter numbers compared to areas with larger pebbles, potentially resulting in lower backscatter intensity. Histogram (e) and (f) show the distribution of VV and VH amplitude value across the 400 m wide section of the river (b) from each 20m² pixel. The darker gravel bars in the east have relatively lower VV amplitude compared to the western gravel beds, and relatively higher VH amplitude, interpreted as due to minor vegetation. © Microsoft**


Figure 3 illustrates the average amplitude cover from January to May 2019 for the study area, from Sentinel-1 A/B SAR ground range detection (GRD) images in descending frame and Interferometric Wide Swath (IW) mode, with a resolution of 25 meters, sourced from Google Earth Engine (https://developers.google.com/earth-engine/datasets/catalog/COPERNICUS_S1_GRD). The SAR amplitude processing steps executed using the Sentinel-1 Toolbox include applying the orbit file, removing border

noise in Ground Range Detected (GRD) format, eliminating thermal noise, applying radiometric calibration values, and correcting terrain geometric distortion. The image presents polarization in VV (average -12 dB along river channels) and VH (average -20 dB along river channels). Notably, the VV polarization amplitude is double that of the VH polarization, which is why VV SAR images are utilized in this study. Additionally, there is no noticeable amplitude change at the GST, indicating that the 6 cm C-band SAR wave does not detect the scale of roughness change from a gravel riverbed to a sandy riverbed with



ripples. Since the amplitude is average in dry season of year 2019, the effect of soil moisture of the sandy riverbed might be less significant.

**Figure 3: (a) Sentinel-1 SAR amplitude in decibels (dB) with VV and VH polarization, sourced from Google Earth Engine. Sentinel-**
**1 is dual-polarized, providing only VV and VH polarized data. The mountain regions show strong backscatter amplitude mainly due to slope orientation. Slopes that face the radar look direction can backscatter more energy return back to the satellite sensor. The forest area has higher amplitude due to strong volume backscattering, but the forest and vegetation cover areas have low coherence due to the seasonal growth change. Along the river channel VV has a higher amplitude than VH. (b) Amplitude value were plotted along river channels and all aligned to start from mountain front, indicates the VV polarization has double the**
**amplitude value compare to VH polarization. The river mountain front is the point of exit of the river from the mountains where the channel abruptly widens. There is no clear amplitude change response to roughness change at the gravel to sand transition along**



**the rivers. The mean amplitude between the pebble and sandy sections of river 2 differs by only 1 dB (Fig. S1). This means that the 6 cm C-band SAR wave could not see the scale of roughness change from gravel riverbed to the sandy riverbed ripples. The SAR amplitude values are plotted here with a 30 m sampling interval. © Google Earth**


## 3.2 SAR interferometric coherence analysis for dry gravel riverbeds

To ensure the reliability of phase difference values calculated from two SAR images, a high coherence value is required (Martone et al., 2012). The coherence in interferometry of two SAR images, taken from the same location but at different times, is determined through their correlation. Coherence is a measure of the similarity between two SAR images' phase

acquired at different times (Goldstein et al., 1988). High coherence of the signal between pairs of images indicates that the radar signals are consistent between the two acquisitions with high correlated phase, while low coherence means the signals are decorrelated. Coherence is an indicator of interferometric phase quality, calculated for each pixel using:

$$coherence = \frac{|S_1 S_2{}^*|}{\sqrt{|S_1|^2|S_2|^2}} \,, \tag{9}$$

In this equation, $S_1$ and $S_2$ represent the complex pixel values at two different acquisition times; S* denotes the complex

conjugate operation of S (Zebker and Villasenor, 1992).

When coherence is 1, it means the backscattering phase is the same, and there is a complete absence of phase noise. The cause of low coherence, which is high phase decorrelation, could be the phase contribution due to high variations in topography, atmospheric phase and orbital error. When two SAR images experience different atmospheric conditions than another (e.g., one SAR image acquired on cloudy day, another SAR image acquired on blue sky day), the difference in the atmospheric delay

can lead to phase noise that result in phase decorrelation (Yu et al., 2018). The phase recorded in a SAR image is highly sensitive to the satellite's position, especially for the topographic phase component and geometric distortion correction. The satellite's position at the time of SAR image acquisition determines the viewing angle, which is crucial information for correcting geometric distortions in SAR images. Fortunately, the Sentinel-1 satellite has well-constrained orbital control with precise orbital recording files. This means that orbital phase errors can be effectively corrected in Sentinel-1 SAR images

(Filipponi, 2019).

For obtaining good InSAR results, especially in applications like monitoring tectonic deformation and topographic mapping, coherence values typically need to be relatively high. A commonly accepted threshold for "good" InSAR results is a coherence value of 0.3 to 0.6, or higher (Cigna and Sowter, 2017). This threshold can vary depending on specific applications and the environmental conditions of the area being studied. High coherence (>0.6) is ideal for most InSAR applications, indicating

strong similarity between the two SAR images. High coherence is essential for detailed deformation analysis or for detecting subtle changes in the Earth's elevation. Moderate coherence (0.3 to 0.6) is still useful, but the results may have higher uncertainty that may be adequate for broad studies where fine details are not as critical. Coherence lower than 0.3 means noise could be more prominent in the phase information. Figure 4 shows 20 m resolution, high coherence value from year 2015-2023 dry season, in red colour with coherence value around 0.8 along the gravel riverbeds.



The Sentinel-1 SAR Single Look Complex (SLC) interferograms were processed in full resolution (~5 x 20 m), then geocoded into 20 m resolution pixel size SAR interferograms (Lazecký et al., 2020). For Sentinel-1 SAR Interferometric Wide Swath (IW) acquisition mode, the single look ground range resolution (across-track direction) is approximately 5 m, the azimuth resolution (along-track direction) is approximately 20 m. The low multi-look value of range 4 and azimuth 1 only works at the high coherence area, which in this case is the dry season gravel riverbeds (Fig. 4). Again, 20 m resolution SAR images are

solely used to map elevation change along dry riverbeds.

In contrast, the SAR images with a 100 m resolution are generated by averaging neighbouring pixels; this averaging reduces speckle noises and improves the signal-to-noise ratio to effectively address the low coherence issue in the cropland. This is sufficient for mapping the background basin elevation change. For 100 m resolution InSAR processing, the input consists of data from all seasons, aimed at mapping the background basin elevation change signal.


**Figure 4: (a) The coherence map with 20 m resolution across the study area during the season; (b) Coherence time-series at the black point in (a) on river 2 shows seasonal variation. Within the same year, the short timespan coherence is higher during the dry season**



**and lower during the monsoon season, probably due to the waves on the river surface causing low coherence. The long timespan interferograms that cross two dry seasons exhibit low coherence, likely due to sediment erosion and deposition during the migration of channels and bar-forms caused by the monsoon floods; (c) The coherence value were plotted along river channels providing insights into the spatial variability with troughs at 0.3 and peaks at 0.8. The coherence troughs are typically found at the edges of the channel and vegetated sand bars, which results in fewer data points in the final InSAR results. These areas of low coherence do not show a noticeable alteration in the trend of InSAR elevation change results, but the data points in the trend are more scattered**
**(Fig. 14). © Google Earth**

### 3.3 Short timespan and long timespan SAR interferometric phase (20 m resolution) analysis for gravel riverbeds

A SAR phase image is expressed in Eq. (10), and SAR interferometric phase is the phase change between two SAR images acquired at the same area from two different times (Flores-Anderson et al., 2019), calculate by Eq. (11). The 30 m resolution
SRTM DEM was used to remove static topography phase during the interferogram calculation (Lazecký et al. 2020) (Fig. 5).

$$phase_1 = arctan\frac{b_1}{a_1} = \emptyset_{displacement1} + \emptyset_{topography1} + \emptyset_{flat1} + \emptyset_{atmosphere1} \ , \tag{10}$$

$$InSAR \ phase = phase \ difference = phase_2 - phase_1 \ , \tag{11}$$

The SAR backscatter phase values are always a mixture of different source of phases, such as flat earth (earth curvature), topography, displacement and atmospheric phase (Hanssen, 2001). After InSAR phase calculation, the residual topographic phase and line-of-sight displacement phase are dominant phase in the most cases (Gaber et al., 2017). There are two Multi-
Temporal Interferometric Synthetic Aperture Radar (MT-InSAR) techniques: Permanent Scatterers (PS) and (Small Baseline Subset) SBAS. The PS-InSAR method estimates the residual topographic phase by using its linear relationship with the perpendicular baseline. This technique models and eliminates residual topographic phases by comparing phase observations across images that have varying baselines, thereby effectively removing the residual topographic phase (Ferretti et al., 2001;
Hooper et al., 2004). The SBAS-InSAR method relies on the accuracy of the DEM used to remove the topographic phase. If the DEM is outdated, it will include residual topographic phase mixed with the LOS motion phase as input for SBAS inversion, leading to higher uncertainty in the LOS displacement results (Berardino et al., 2002; Morishita et al., 2020). The PS-InSAR method relies on persistent backscatters, making it most effective in urban areas with strong double bounce scattering. Conversely, the SBAS-InSAR method relies on distributed backscatters (one phase value from the sum of a pixel's
backscatters), particularly effective in areas with lower coherence and mixed backscatter mechanisms types (diffuse scattering, volume scattering and double-bounce). In this study, we have selected the SBAS-InSAR method as it is better suited for rural areas with diffuse scattering (Fig. 2).

The short timespan interferograms refer to time pairs are over shorter intervals than 90 days, as InSAR coherence typically exhibits seasonal variations in the study area. Long timespan interferograms span 90 to 360 days, designed to bridge network
gaps across different seasons. The short timespan interferogram from the dry season maintains good coherence, with an average value of 0.6 along dry riverbeds, ensuring reliable interferometric phase calculations (Fig. 6); this would be expected if there is little disturbance of the riverbeds between the time pairs. The interferograms are unwrapped using a statistical cost approach





with the SNAPHU software (Chen and Zebker, 2002). The long timespan interferograms exhibit low coherence, probably linked to sediment erosion and deposition during the migration of channels and bar-forms during the monsoon floods (Fig. 7).

These decorrelated phases are insufficient for obtaining reliable interferograms, and the long timespan interferograms are excluded from the 20 m resolution SBAS-InSAR processing.

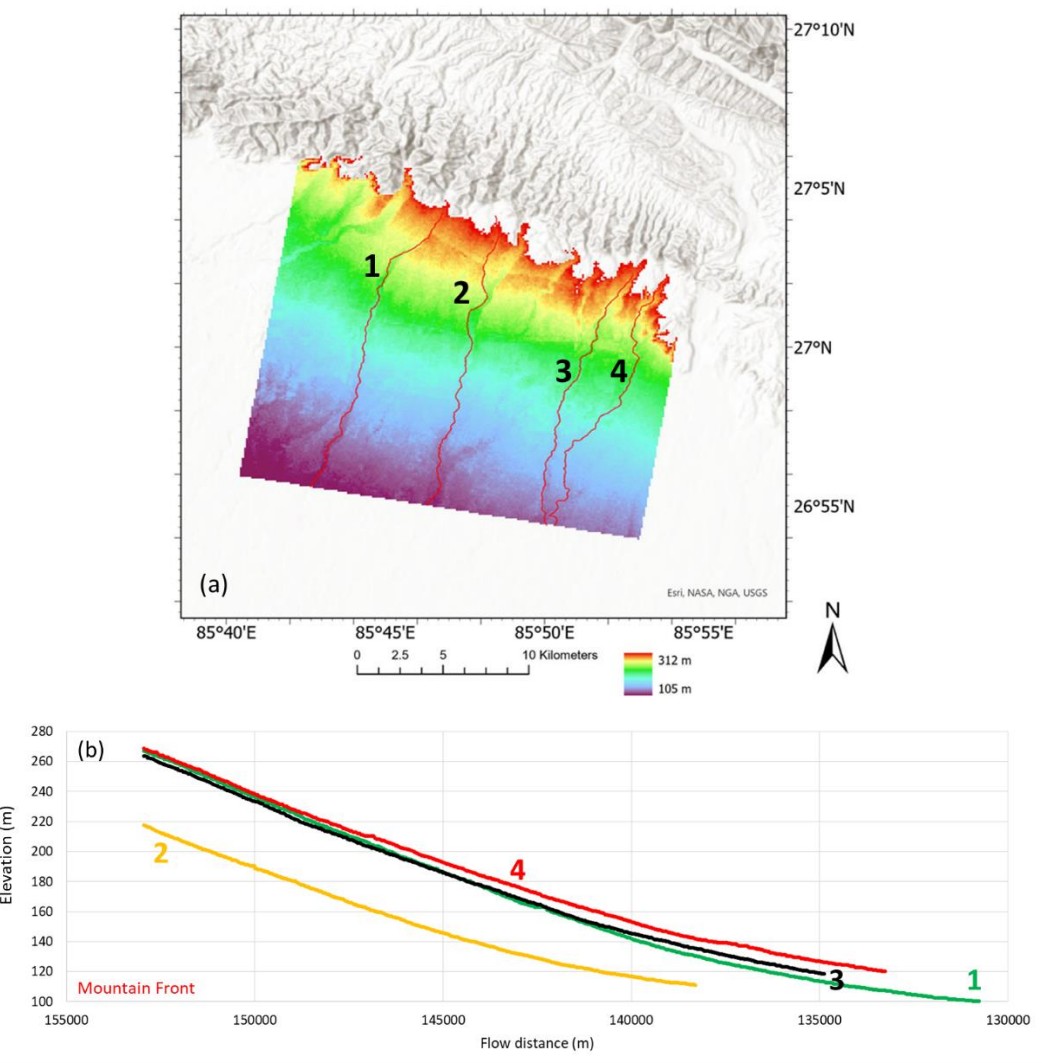

**Figure 5: (a) The 30 m resolution SRTM DEM. The terrain results in a gradient in the topographic phase values. This DEM was**
**used to remove static topography phase during the interferogram calculation process, which is carried out using the GAMMA software as implemented by the LiCSAR (Lazecký et al., 2020). The residual topographic phase is derived by subtracting the topographic phase from the year 2000 SRTM DEM. (b) River long profiles showing no evidence of knick points. Along the river from the mountain front to the gravel-sand transition, the elevation decreases by around 150 m down slope. The exact season when the DEM data was collected is unknown. Consequently, the elevation data along the river channel may represent the water surface**
**elevation rather than the dry riverbeds elevation.**



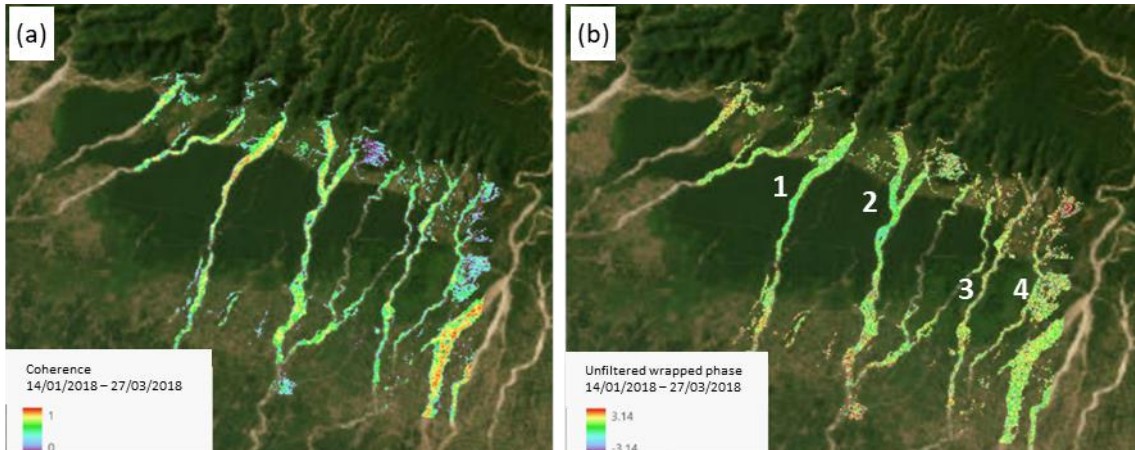

**Figure 6: Short timespan (acquired on dates 14/01/2018 and 27/03/2018, 72 days) (a) coherence and (b) unfiltered wrapped phase interferogram (20 m resolution, 0.2 coherence masked). Phase changes between adjacent pixels is relatively small and continuous. The continuous shifts in colour along the river shows a smooth, low phase gradient feature that contributes to a more accurate phase unwrapped interferogram. © Google Earth**

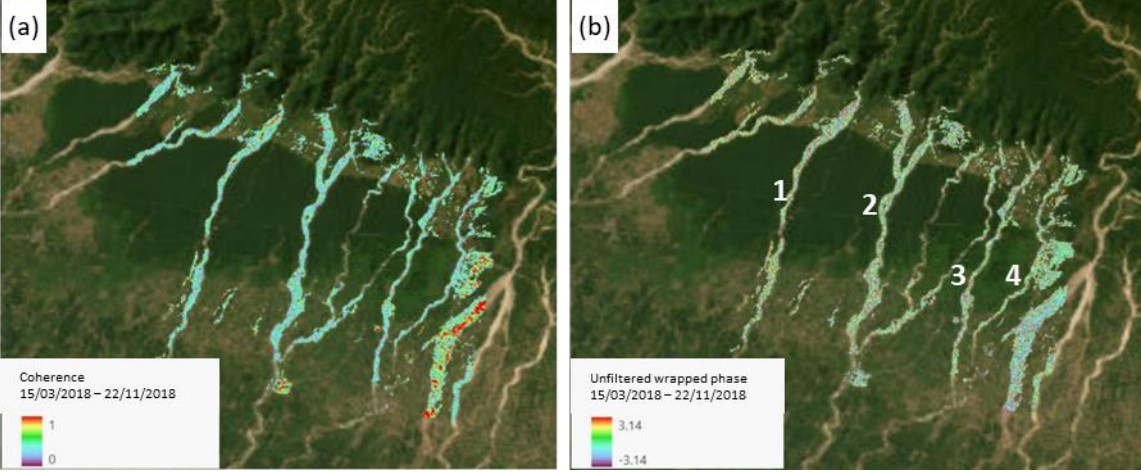

**Figure 7: Long timespan (acquired on dates 15/03/2018 and 22/11/2018, 252 days) (a) coherence and (b) unfiltered wrapped phase interferogram (b) (20 m resolution, 0.2 coherence masked). Based on coherence data, the months from November to March are identified as the driest in the study area. Most regions shown in blue on (a) display coherence values below 0.3, indicating that they are insufficient for obtaining reliable interferometric phase. Areas of high coherence on the eastern side are depicted in red on (a) corresponds to an embanked, inactive gravel riverbed. © Google Earth**

## 4 Methods and data applied in this study

### 4.1 InSAR data and its network for SBAS-InSAR processing

The SBAS (Small Baseline Subset) method for InSAR (Interferometric Synthetic Aperture Radar) processing was developed in the early 2000s by Berardino et al. (2002) to invert the temporal surface displacement. It is a sophisticated and common



method, and in this study we implement the SBAS-InSAR method by using LiCSBAS software (Morishita et al., 2020). The

atmospheric noise correction is applied by using the GACOS data (Yu et al., 2018). The success of this method heavily relies

on the quality and availability of abundant InSAR images.

We processed two SBAS-InSAR datasets aimed at monitoring changes in both channel and floodplain elevation changes: 20

m resolution data specific to the dry season (October – May) from year 2016 to 2021 targeted the channels (Fig. 8), and 100

m resolution covering all seasons from year 2016 to 2021 aimed at monitoring the floodplains (Fig. 9). Reference point

selection is another important factor, due to having limited coverage area, a relatively stable reference point for both resolution

processing is important. We choose to use a reference point at an airport (85.86 E 26.93N) in an embanked not active gravel

riverbed, marked in Fig. 10 and 11.

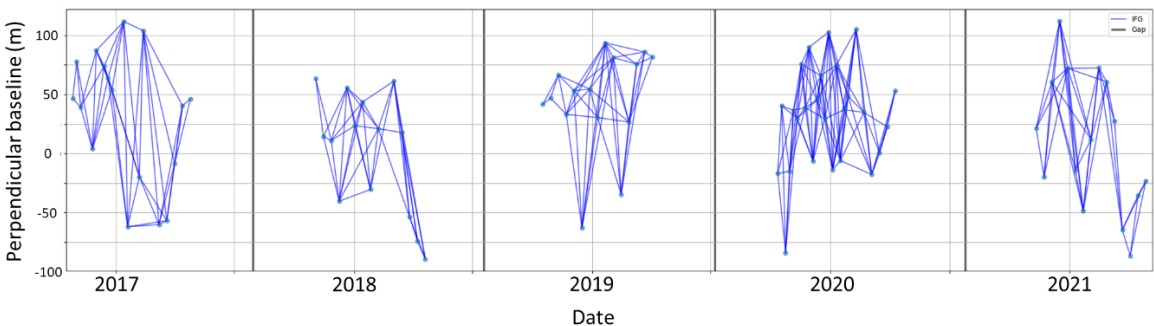

**Figure 8: (a) A network of 20 m resolution interferograms. The y-axis is the perpendicular baseline, versus acquisition dates on the**
**x-axis. The perpendicular baseline is the distance between the satellite orbits when the satellite revisits the 'same location'. Each blue**
**line connects two SAR images for calculating interferograms. We are leveraging the residual topographic phase, so the gaps in the**
**network is 'filled' by the differential residual topographic phase. Maintaining a consistent range of perpendicular baselines within**
**each network segment is crucial for preserving the sensitivity to topographic phase changes (Fattahi and Amelung 2013).**

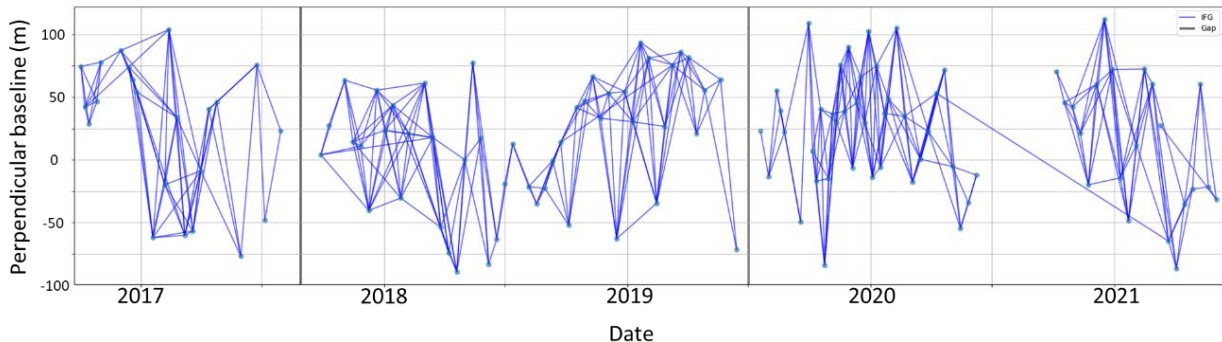

**Figure 9: A network of 100 m resolution interferograms for mapping the basin cropland. This is based on the traditional LOS**
**displacement phase, so the gaps are filled by the good quality long timespan interferogram. It's important to highlight that long**
**timespan interferograms with a 20 m resolution are not effective for analysing gravel riverbeds. The 100 m resolution long timespan**
**interferogram is sufficient for croplands mapping, offering a practical solution to bridge the network gaps. There are two small time**
**gaps around in the 100 m resolution network, linked by linear fit through LiCSBAS processing. Due to the gaps being relatively**
**small, there is an insignificant effect on the subsidence time-series analysis in Section 4.**





## 4.2 SBAS-InSAR processing based on differential residual topographic phase

Along dry gravel riverbeds, the phase change sensitivity to the perpendicular baseline (Fig. 8), indicates that the phase is dominated by the residual topographic phase. A previous study (Du et al., 2016) demonstrated that utilizing the MT-InSAR
technique, which is based on stacked interferograms, and enhances the accuracy of calculating the residual topographic phase. They also observed the sensitivity of residual topographic phase is influenced by the overall range of perpendicular baselines within the entire network of interferograms for each time segment, rather than the baseline of any single interferogram in the stacked calculation. The stability of residual topographic phase estimation is maintained, irrespective of the way in which the selected short-baseline interferograms are linked (Du et al., 2016; Fattahi and Amelung, 2016).

The residual topographic phase has traditionally been treated as noise to be removed for accurate line-of-sight displacement phase processing. However, our study reveals that the dry gravel riverbeds at the Himalayan mountain front provide favourable geomorphic setting for retrieving high quality residual topographic phases. This is attributed to their annual bedload sedimentation and the strong diffuse scattering caused by the gravel. Additionally, the Sentinel-1 SAR data, with its 10-year span, 12-day revisit frequency, and well-constrained orbital tube, offers a rich dataset for recording these accurate residual
topographic phases.

In our study, we use the SBAS-InSAR technique to calculate the annual residual topographic phase. We then fit a linear model across the residual topographic phase across different years. In our study, we maintained the perpendicular baseline within a range of ±100 meters because the majority of the interferograms fall within this range. The differential residual topographic phase values are then converted to annual elevation change rates using Eq. (15). This procedure is implemented using the
LiCSBAS code (Morishita et al., 2020).

To summarize, the SBAS-InSAR processing based on differential residual topographic phase relies on several assumptions: (1) The residual topographic phase is the predominant phase value along the dry riverbeds, unaffected by noise and line-of-sight (LOS) displacement phase. To support this assumption, the background LOS displacement signal must be analyzed and separated. The time-series mapping of the basin background indicates that the LOS displacement remains 'flat' during the dry
season (Fig. 18). Therefore, we assume that the phase observed along the dry gravel riverbeds is primarily from the residual topographic phase. Additionally, we examined the unwrapped phase profile along the river and its sensitivity to the perpendicular baseline, which demonstrates a positive linear relationship between topographic phase sensitivity and the perpendicular baseline (Fig. 10); (2) The network connectivity of each acquisition time results in similar topographic phase sensitivity (phase ambiguity), as indicated by relatively flat time-series within the same year (Fig. 16); (3) We account for
variations in the scaling factor by calculating the average perpendicular baselines for the five different connected networks are 52.2 m, 52.8 m, 49.5 m, 48.4 m, and 50.4 m (Fig. 8). Consequently, the ratios of $B_{\perp 2}/B_{\perp 1}$ are 1.01, 0.94, 0.94, and 1.04. To quantify the uncertainty percentage caused by these ratios, we conducted forward modelling (Supplements 2) and observed their effect on the elevation change ratios to be +2%, -12%, -12%, and +8%. Therefore, we conclude that the impact of the scaling factor on the final result's uncertainty percentage falls within the range of +8% to -12% (Fig. 14).



$$\emptyset_{residual\_topo1} = \frac{4\pi B_{\perp 1}}{\lambda} \frac{H_1 - H_{dem}}{R_1 \sin(\theta)} , \qquad (12)$$

$$\emptyset_{residual\_topo2} = \frac{4\pi B_{\perp 2}}{\lambda} \frac{H_2 - H_{dem}}{R_2 \sin(\theta)} , \qquad (13)$$

$$\Delta\emptyset_{residual\_topo} = \frac{4\pi}{\lambda \sin(\theta)} \left( \frac{B_{\perp 2} (H_2 - H_{dem})}{R_2} - \frac{B_{\perp 1} (H_1 - H_{dem})}{R_1} \right) , \qquad (14)$$

After applying the parallel-ray approximation, the mathematical relationship between residual topographic phase $\Phi_{residual\_topo}$ and height (H) can be written in Eq. (12) (Fattahi and Amelung, 2013; Pepe and Calò, 2017). By combining Eq. (12) and (13), we are able to derive Eq. (14), where $B_{\perp}$ is the perpendicular baseline, $\lambda$ is SAR wavelength, $\Theta$ is the satellite SAR acquisition average incident angle, R is distance between satellite and earth surface (Fig. S2). In Eq. (14), the parameter R is approximately 700 km. Since centimetre level surface displacements are negligible in comparison to R, it is reasonable to assume that $R_1 = R_2$. Then we could write Eq. (14) as following,

$$\Delta\emptyset_{residual\_topo} = \frac{4\pi B_{\perp 1}}{\lambda \sin(\theta)R} \left( \frac{B_{\perp 2}}{B_{\perp 1}} (H_2 - H_{dem}) - (H_1 - H_{dem}) \right) , \qquad (15)$$

Based on Eq. (15), the key component of the red-coloured scaling factor is the ratio $B_{\perp 2}/B_{\perp 1}$, which influences the percentage of the elevation change results. The remaining terms are constants and have no impact on the elevation change results. Ideally, if $B_{\perp 2}/B_{\perp 1}=1$, we would achieve the perfect elevation change results. During data processing, the goal is to balance the number of input interferograms while keeping the ratio of $B_{\perp 2}/B_{\perp 1}$ as close to 1 as possible.

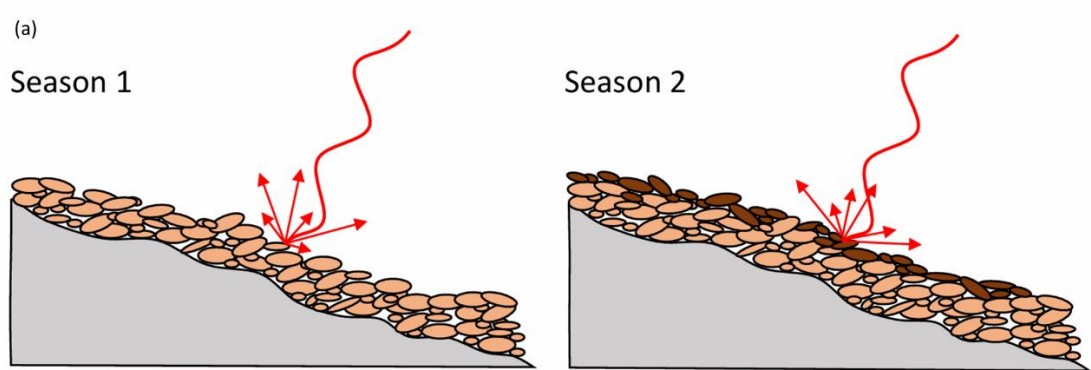



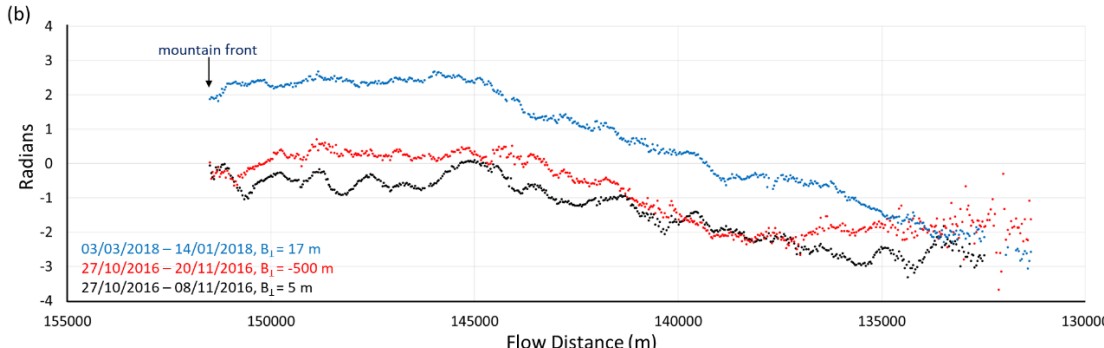

**Figure 10: (a)** Illustration of the annual increase in dry gravel riverbed topographic elevation due to sediment aggradation. Dark brown pebbles show accumulation in season 2. **(b)** Changes in the residual topographic phase (in radians) along river 2, influenced by sediment aggradation and perpendicular baseline variations. The filtered unwrapped phase values are plotted along river 2, comparing residual topographic phase observations with different perpendicular baselines from the same year (black and red curves), and the residual topographic differences between 2016 and 2018 with similar perpendicular baseline (blue and black curves). From the same year, the large baseline (red) measured topographic phases are higher than smaller baseline measurements (black). This indicates that a larger perpendicular baseline increases the sensitivity of the interferometric phase to topographic elevation. Comparing different years, the year 2018 (blue) has higher residual topographic phase values, probably due to sediment aggradation.

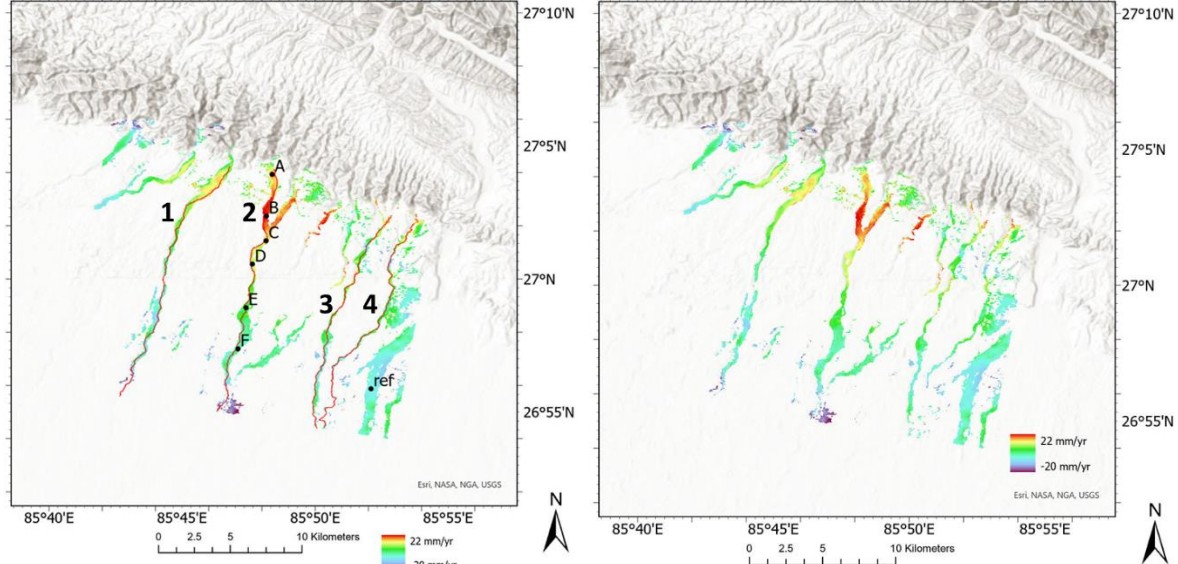

**Figure 11:** Spatial distribution of topographic change in 20 m resolution 2016 – 2021 for dry gravel riverbeds map with and without annotation. The results show the positive elevation change gradually decreasing with distance from the mountain front. The 20 m resolution SAR dataset focuses solely on mapping the rate of elevation change of dry river channels based on the residual topographic phase. Basemap data sources: ESRI World Topographic basemap with hillshade illumination.



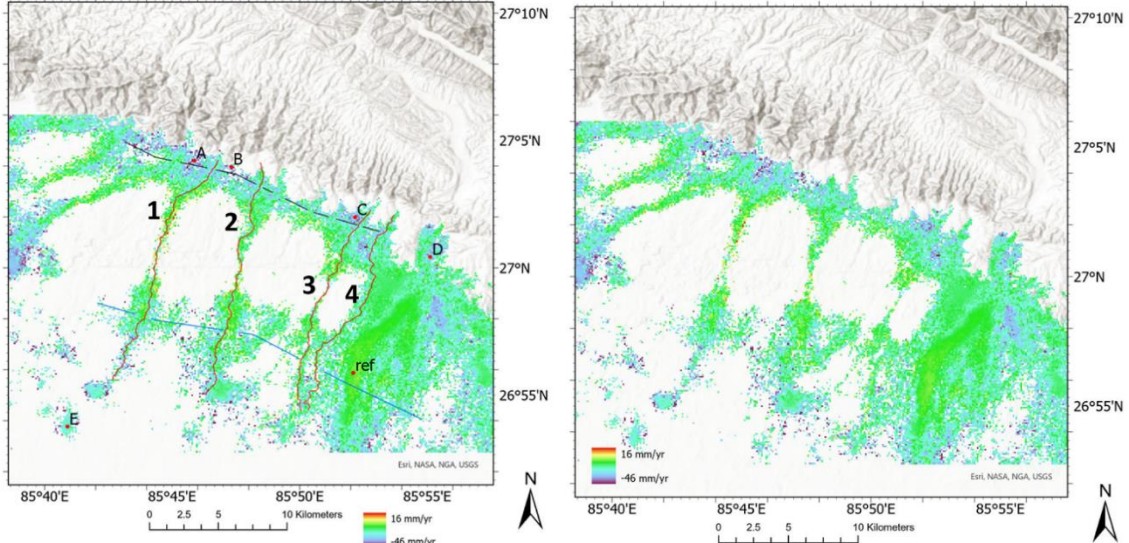

**Figure 12: Spatial distribution of line-of-sight (LOS) displacement in 100 m resolution from 2016 – 2021 for background basin elevation change map with and without annotation. This figure shows the negative elevation change in the basin. Due to the vegetation decorrelation effect, the results are quite sparse. The villages have a denser pixels, attributable to the strong double bounce backscattering caused by houses. The 100 m resolution result is only for mapping the basin based on line-of-sight displacement phase. For detailed LiCSBAS processing implementation based on LOS displacement phase, please see the appendix. At the river, the data are considered unreliable due to contamination from monsoon season river water. Basemap data sources: ESRI World Topographic basemap with hillshade illumination.**

## 4.3 SBAS-InSAR velocity standard deviation

The phase change related elevation change is sensitive to sub-wavelength elevation change, which is the basis for the InSAR technique. This is based on the condition of reliable phase differences as input for the data processing. Accurately measuring the phase difference is crucial, and efforts should be made to minimize noise. Typically, this phase is the Line-of-Sight (LOS) displacement phase. However, in our 20 m resolution processing for dry gravel riverbeds, it is based on the differential residual topographic phase. This does not change the character of sensitive to sub-wavelength elevation change. Figure 13 displays the standard deviation of the final InSAR velocity. A higher standard deviation indicates greater variability and noise in the measured velocity. The standard deviation is primarily influenced by the quality of the unwrapped interferogram used to calculate the final InSAR velocity. The velocity standard deviation is calculated based on a method called the bootstrap, which uses the cumulative displacement data and repeated bootstrap sampling from original cumulative displacement data, then calculate the velocity. The standard deviation of the velocity is shown in Figure 14, which tells us how much the velocity estimate vary. Note that the standard deviation might be underestimated if the network is not fully connected, due to the temporal constraint in the small baseline inversion (Morishita et al., 2020).



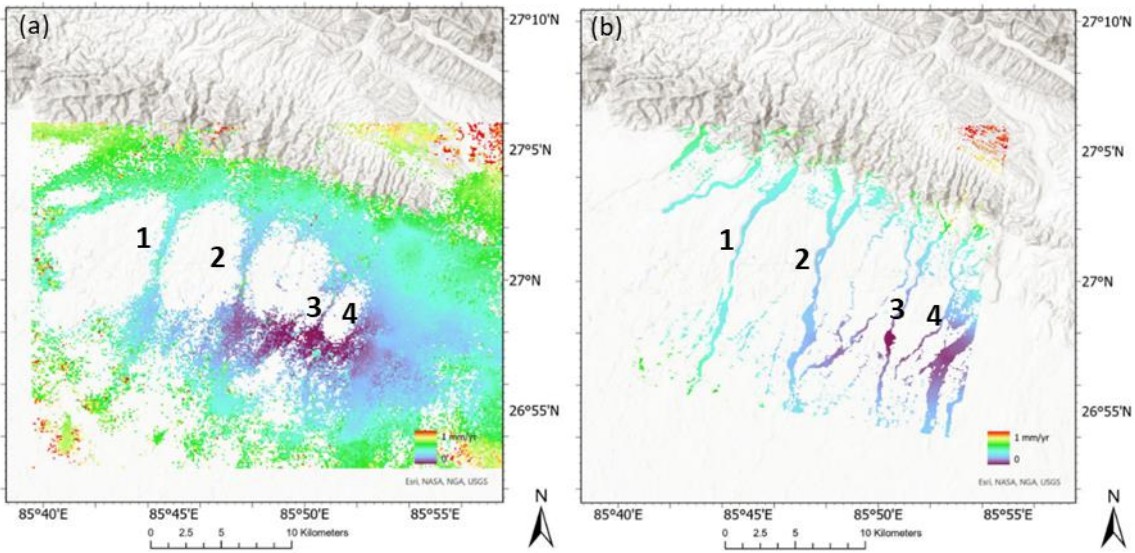

**Figure 13: InSAR LOS velocity for 100 m resolution (a) and 20 m resolution (b) standard deviation ranges 0–1 mm/yr. The**
**calculation of the velocity standard deviation assumes that all input pixel values from the InSAR images are reliable. For the 100 m**
**resolution InSAR results at river channels, despite showing a positive elevation trend, the values are considered unreliable due to**
**high noise contamination. This unreliability comes from noise in the InSAR images taken during the monsoon season. Consequently,**
**the 100 m resolution InSAR results are solely employed for mapping basin elevation changes, while the 20 m resolution InSAR**
**results are used to map changes in river channel elevation. Basemap data sources: ESRI World Topographic basemap with hillshade**
**illumination.**

## 5 Results

### 5.1 InSAR signals of fluvial elevation change

A spatial distribution map from the 20 m InSAR result analysis along rivers 1-4 indicates a positive elevation change along
the river channels. Near the mountain front, the upstream section of the rivers experiences a positive elevation change of
approximately 20 mm from one dry winter season to the next (i.e., 20 mm/yr), which gradually decreases with distance from
the mountain front (Fig. 14). Initially, all rivers exhibit a decline in the rate of elevation change that ends at the forest's edge.
The elevation change progressively declines through the forest to a minimum of about 5 mm/yr. Beyond this, the elevation
change varies between 0 and 5 mm/yr until it reaches the gravel-sand transition, where it begins to fall below zero (Fig. S4).






**Figure 14: InSAR results along the riverbeds show varying vertical elevation change rates: river 1 peaks at 15 mm/yr, river 2 at 25 mm/yr, river 3 at 20 mm/yr, and river 4 at 15 mm/yr. The uncertainty range due to the scaling factor effect, shaded in grey, spans from -12% to +8%. The plot uses a vertical exaggeration of 250,000, meaning the vertical scale is magnified 250,000 times relative to the horizontal. Each dot represents elevation change rates over a 20 m² pixel along the dry riverbeds.**



## 5.2 InSAR signals of non-fluvial elevation change

A spatial distribution map from the 100 m resolution InSAR result analysis in the basin indicates a negative elevation change of the surrounding floodplains. Figure 15 displays two plotted cross-sections: one situated north of the forest (marked with a black-coloured dot) and the other located south of the forest (marked with a blue-coloured dot). Both transect show negative
elevation change. The southern (blue) transect has values between 0 and -15 mm/yr, while the northern (black) transect has values between -5 and -25 mm/yr. The elevation change along river channels are excluded, due to the high noise contamination from SAR images captured during the monsoon season. The transect focuses on illustrating the non-fluvial elevation changes within the basin.

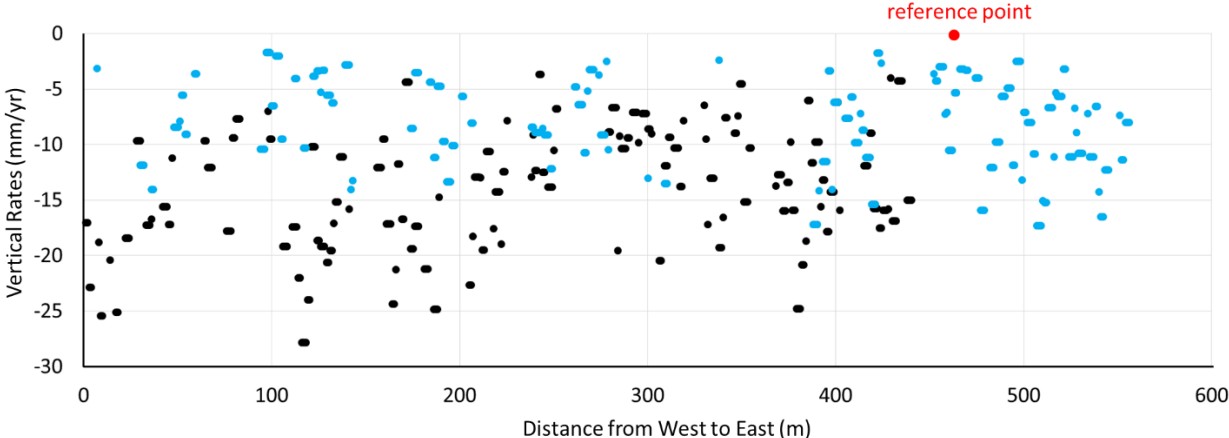

**Figure 15: InSAR result basin transects analysis (black and blue lines in Fig. 12) indicate high heterogeneity in subsidence distribution across the basin, potentially caused by water extraction. Values from cropland and villages are shown in the transect plot. Elevation changes along the riverbeds, affected by monsoon season interferograms, are excluded from the transect plot. The reference point for InSAR processing, located near the blue transect, is projected onto it and marked with a red dot.**

## 6 Temporal analysis of elevation change

Based on the character of the time-series pattern, we can interpret the cause of the elevation change. We chose 6 different locations (dots in Fig. 12) that characterize the typical signal pattern of the time-series along rivers with a 20 m resolution. The locations analysed in further detail are located along river 2, with minimum coherence value 0.5, and maximum velocity standard deviation 0.5 mm/yr (Fig. 12). Based on the time-series patterns observed at six locations along river 2, the characteristics are similar, showing minor fluctuations around their average values, followed by an increase in elevation after
the monsoon (Fig. 16). These fluctuations are during the dry period where no water is discharged through these channels. The reason for the fluctuation in the measurement points (Fig. 16) could be a mixture of noise and the diversity of perpendicular baselines, which caused different topographic sensitivity. However, the fluctuation is within a 5 mm range and did not cause a dramatic change in vertical displacement measurement; it is still within the range of a 'flat' feature. We interpret that the





inter-seasonal displacements shown in the time series are predominantly due to the differential residual topography phase
caused by sediment aggradation along river 2.

We chose 5 different locations (dots in Fig. 12) that characterize the typical signal pattern of the time-series in the floodplain cropland areas in 100 m resolution results. The locations are spread in high coherence pixels in cropland and villages, with minimum coherence value 0.4, and maximum velocity standard deviation 0.5 mm/yr. Based on the time-series patterns observed at five locations in the basin, the characteristics are similar, with a meaningful subsiding trend between each point
during the crop grown season (Fig. 18).

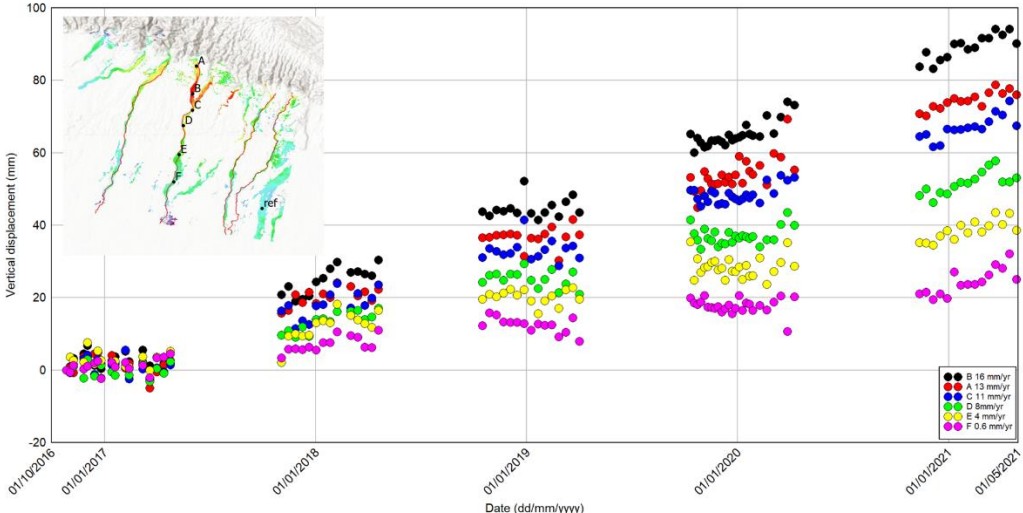

**Figure 16: The SBAS-InSAR (20 m resolution) time-series along river 2 shows positive elevation changes. The June to September gap corresponds to the monsoon season. The inter-seasonal displacements shown in the time series are predominantly due to sediment aggradation. The fluctuations in the measurement points during the dry period could be due to a combination of noise and**
**varying perpendicular baselines caused topographic sensitivity variation. Basemap data sources: ESRI World Topographic basemap with hillshade illumination.**

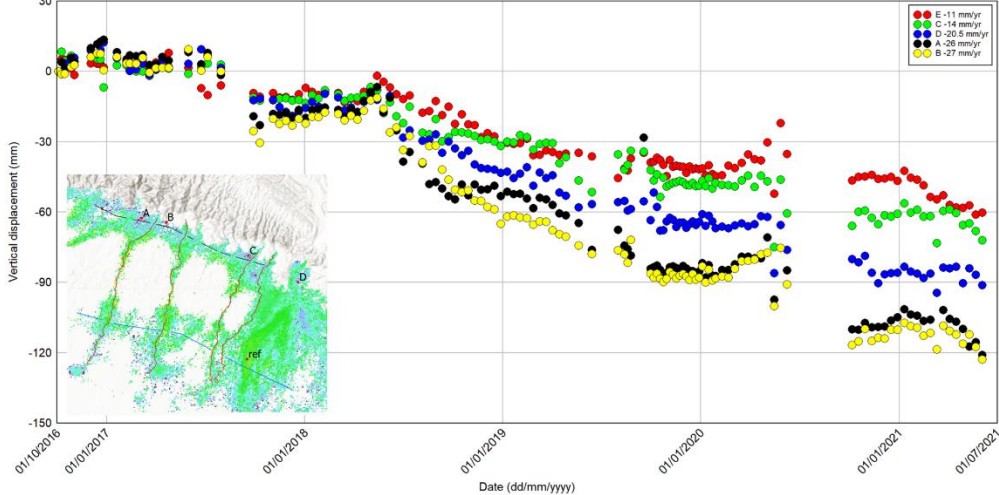





**Figure 17: The SBAS-InSAR (100 m resolution) time-series in cropland shows a consistent pattern of negative elevation change, indicative of subsidence at each observed location. To ensure accuracy, there are no gaps in the network, which is connected by both short and long timespan interferograms. The pattern remains flat from November to May and then trends downward from June to October. Basemap data sources: ESRI World Topographic basemap with hillshade illumination.**

## 7 Uncertainties generated by working in active river settings

To map the river aggradation rate, satellite DEMs offer meter-scale accuracy, while airborne LiDAR can achieve accuracies within tens of centimeters. Two-pass InSAR (DInSAR) typically maps elevation changes with accuracy up to a few centimeters, but MT-InSAR processing can improve the accuracy to millimeter scale. These are general ranges, and the actual accuracy can vary by project. In this study, we enhance elevation mapping accuracy from meters to millimeters by leveraging the residual topographic phase value combined with MT-InSAR processing techniques, rather than relying on absolute topographic phase values for DEM mapping.

The key to effectively using the average residual topographic phase for mapping elevation changes rests in confirming whether the feature of elevation change supports the precise measurement of the topographic phase. In our study, we observed that the four rivers completely dry out during the dry season, leaving the surface undisturbed. Most importantly, the 100 m resolution basin mapping indicating minimal Line-of-Sight (LOS) displacement phase during dry season (Fig. 18). This observation is crucial as it ensures that the dominant phase is due to the residual topographic phase, rather than LOS displacement phase. In our study area, the LOS displacement phase difference is positive, while the differential residual topography phase causes a negative phase difference. They contribute in opposite values to the final phase difference value, affecting the overall phase value measurement. Therefore, it is crucial to have a predominantly topographic phase mapping time period, with minimal LOS displacement phase component mixed in the observed phase.

We have a seven-month dry season window to acquire abundant SAR images for mapping topographic phase. Maintaining a sufficient number of interferograms for SBAS-InSAR analysis input is important to ensure reliable and stable results. Using data from only two years leads to results with high uncertainties. Therefore, we have incorporated five years of Sentinel-1 data, with a minimum of 15 interferograms each year. In our study area, the low phase gradients facilitate accurate phase unwrapping. Additionally, conditions such as low atmospheric phase noise during the dry season, and the absence of layover, shadow, and foreshortening geometry distortions, contribute favourably to get the accurate topographic phase value.

The sensitivity of the topographic phase has a linear relationship with the perpendicular baseline. Therefore, maintaining a consistent range of perpendicular baselines within each network segment is essential to ensure a consistent sensitivity of topographic phase measurement (Fattahi and Amelung, 2013). Sentinel-1's precise and stable orbital tube plays a crucial role in baseline control for this study. We have maintained the perpendicular baseline within a range of ±100 meters because the majority of the interferogram is within this baseline range (Fig. 8). As outlined in Eq. (14), the LOS displacement between each year must also be minimal to support the assumption that the scaling factor is '1'. It is important to note that the scaling factor does not influence the trend of elevation change.



## 8 Discussion

### 8.1 Validation of the SBAS-InSAR method in active river setting

Direct validation of the SBAS-InSAR derived approximations of sediment aggradation rates requires field-based monitoring
at a temporal (annual) and spatial scale (20 m² pixel area) that is comparable to the SBAS-InSAR monitoring. This is a priority
for follow-on research, and may be achieved using repeat drone mounted LiDAR surveys calibrated using differential GPS
surveys during the dry seasons (Wheaton et al., 2010; Williams, 2012). A similar approach may use Satellite LiDAR, such as
the Ice, Cloud, and Land Elevation Satellite (ICESat), which has a 10 m diameter pixel size and a vertical elevation change
accuracy of 30 mm/yr (Schutz et al., 2005). Although the vertical elevation change accuracy of SBAS-InSAR is higher, at 1
mm/yr, satellite LiDAR would still be able to test the first order signals of elevation change. As yet, no publications have
reported on ICESat measurements of river aggradation rates at the Himalayan Mountain Front.

Indirect validation of the SBAS-InSAR derived approximations of sediment aggradation rates may be made by comparison to
other measurements of sediment aggradation rates in channels of the Gangetic Plains. For example, Sinha et al. (2019, 2023)
report sediment aggradation rates in the lower Ganga River ranging from approximately 10 to 90 mm/yr, and in the Kosi River
between 40 and 50 mm/yr, based on sediment load measurements. Floodplain sedimentation rates for the upper Yamuna Valley
have been measured using 210Pb dating at between ~25 to 60 mm/yr (Saxena et al., 2002). Our measured sediment aggradation
rate at the mountain front is approximately 20 mm/yr, which is smaller than the rates for the much larger Ganga and Kosi
Rivers which intuitively seems reasonable.

### 8.2 Factors contributing to floodplain subsidence and dry gravel riverbeds aggradation rates

In this study area, the interferogram phase obtained along a dry gravel riverbed has an annual increase in topography with no
or minor LOS displacement during dry season. This geomorphic setting favours accurate topographic phase mapping. The
dominant changes in elevation in this region are interpreted to be the result of a combination of slow regional subsidence
driven by tectonics and compaction countered by sediment aggradation in river channels driving increased elevations at rates
of up to ~20 mm/yr. The subsidence of pro-foreland basins such as the Gangetic Plains are usually less than 0.5 mm/yr (Sinclair
and Naylor, 2012; Sinha et al., 2007), and so the observed regional rates (Fig. 12) are faster. Probable reasons include shallow
compaction of sediment and localised anthropogenic water extraction as suggested by the high variance in the rates. All four
rivers show a decline in aggradation rates near the mountain front that coincides with the strip of agricultural land between the
mountain front and the forest. This localised signal diminishes in the forested areas (Fig. 14). The locations of subsidence
coincide with land features, indicating that it may be caused by water extraction for irrigation (Raju et al., 2022) and/or changes
in surface soil moisture (De Zan et al., 2015; Maghsoudi et al., 2022; Zheng and Fattahi, 2022; Wig et al., 2024).

There are three primary factors that influence the strength of SAR backscatter energy from diffuse scattering: surface
roughness, slope, and dielectric properties. Thus, soil moisture (as a dielectric property) is the main factor influencing what
we 'see' from InSAR. The SBAS-InSAR (100 m resolution) time-series across croplands shows the downward trend aligns



with the monsoon season, while the flat trend corresponds to the dry season. In Nepal, the primary crop growing season, from

July to October, coincides with the most pronounced subsidence trends observed in the time series. One interpretation is that the croplands high soil moisture during the monsoon season cause an exaggerated subsidence signal from the actual subsidence value, which is the typical effect of the soil moisture (De Zan and Gomba, 2018; Zheng and Fattahi, 2022). However, a detailed analysis of soil moisture with its seasonal SAR amplitude and phase variation in the cropland area is not the focus of this study, but will be addressed in the future research.

The forested area through which the rivers flow is characterised by slightly higher surface elevations than the surrounding plains suggesting that there may be a long-term background signal of tectonically driven surface uplift that is not recorded through our period of study. This may be generated by a buried thrust tip within the foreland basin that is episodically active and may punctuate the background subsidence rates recorded here.

## 8.3 River dynamics and its avulsion cycles

The sediment aggradation within the channels decreases from the mountain front to the gravel to sand transition. The rates at the mountain front are equivalent to the accumulation of the D50 grainsize across the channel; the decreases are likely to be associated with a slight decrease in grainsize downstream to the gravel-sand transition, although this hasn't been demonstrated in these locations. An implication of these results is that the river channel at the mountain front is slowly increasing in channel gradient and that it is also becoming elevated above the surrounding floodplain. If we consider 'super-elevation' to require the

riverbed to be above the height of the surrounding floodplain (Slingerland and Smith, 2004), and consider the average bankful depth of these rivers to be around 2-5 m, then we would expect aggradation to result in channel avulsion every few hundreds of years (i.e. channel depth divided by aggradation rate).

To predict which river is approaching its next avulsion cycle, we hypothesise that rivers with higher sediment aggradation rates, are more likely to have recently avulsed and are more transient in terms of the transport capacity of the river versus its

gradient. If this is correct, rapidly aggrading channels should exhibit low elevation contrast relative to its floodplain. Conversely, rivers with lower sediment aggradation rates might be 'older' channels nearing their time for an avulsion. In this study, we observed that river 2 had the highest aggradation rate among the four rivers, it also had the lowest elevation compared to the other rivers (Fig. 5 and Fig. 14). This suggests that river 2 maybe a recently avulsed river.

## 8.4 Qualitative analysis of sediment yield on gravel riverbeds

The documentation of sediment aggradation along a channel enables an approximation of the volume of sediment that accumulates in that portion of the channel during a single monsoon. By combining the values for each pixel in river 1 we obtain a total volume flux of ~ 45000 $m^3$/yr. This represents the accumulation of coarse bedload that will be a portion of the total sediment load that was transported through the channel during that period. The upstream catchment that is the source of the sediment has an area of 29,000,000 $m^2$, hence the bedload alone represents an average erosion rate over the catchment of

~1.5 mm/yr. The maximum likely erosion for the catchment is likely to be <5 mm/yr based on known erosion rate



measurements in similar Siwalik Hill settings along strike (Mandal et al., 2023), so it seems likely that the portion of bedload in this setting is likely to be as much as a third of the total flux. This is high, but similar to values obtained from other studies in the Himalaya (Pratt-Sitaula et al., 2007), and is likely a response to the high proportion of Upper Siwalik conglomerates in the section (Dhital, 2015; Pradhan et al., 2004; Pradhan et al., 2005).

**8.5 Future prospects with SAR application in rivers**

The advent of micro-SAR and the anticipated NISAR (NASA-ISRO Synthetic Aperture Radar) missions, providing X-band and S-band SAR images, are expected to provide deeper insights into river aggradation rates and enable more detailed mapping of background floodplain subsidence. Furthermore, the gathering of SAR data over many years is necessary for demonstrating the potential of InSAR time-series analysis. The current background study for retrieving accurate residual topographic phase
is limited. However, we believe that our high accuracy riverbeds mapping results will increase awareness of leveraging residual topographic phase for elevation change mapping. The development of this novel approach is based on differential residual topographic phase and carries several assumptions. We assume the InSAR phase along the dry gravel riverbed is purely residual topographic phase and the scaling factor is 1. Separating the baseline-dependent phase component (topographic phase) from the line-of-sight displacement phase could be considered in more complex situations, such as strong background subsidence
signals in the floodplain during the dry season. Methods such as Persistent Scatterer (PS-InSAR) could be used to implement for separating the phases (Ferretti et al., 2001; Hooper et al., 2004). To achieve accurate topographic phase measured elevation changes, the 'cleaned' residual topographic phase needs to be incorporated back into the SBAS-InSAR technique for elevation change calculation. Additionally, Future investigations should also focus on assessing the precision of retrieving the residual topographic phase across diverse river geomorphologies and under varying land conditions globally. Such research will help
validate the robustness and scalability of this novel approach for its operational potential in developing its use as a standard tool in geomorphic and hydrological research worldwide. Looking ahead SAR remote sensing will likely become standard practice for monitoring ephemeral river aggradation rates, particularly for rivers in the Himalayan Mountain Front region.

**9 Conclusions**

The study underscores the potential of SAR, PolSAR and InSAR technologies in providing detailed, high-resolution
geomorphological data, which is essential for understanding and predicting river dynamics. The dry gravel riverbeds exhibited medium-range backscatter (-12 dB) in VV polarized SAR images and maintained high coherence levels (0.6). The methodology and results presented here offer valuable insights for future research and practical applications in flood risk assessment, sediment management, and land use planning. By highlighting the interplay between sediment dynamics and human activities, this work contributes to a more comprehensive understanding of ephemeral river behaviour in densely
populated regions.

This study demonstrates the effectiveness of applying the Small Baseline Subset (SBAS) InSAR method on residual topographic phase. We successfully mapped millimeter-scale elevation changes in river channel over a ~15 km stretch from the mountain front to the gravel-sand transition in southeastern Nepal. Results indicate significant sediment aggradation in river channels, with rates reaching up to approximately 20 mm/yr near the mountain front, declining downstream. Meanwhile, the floodplain in the basin is subsiding at a rate of around -15 mm/yr. This sediment build-up plays a critical role in increasing the risk of river avulsion, which can have severe implications for flood management and the safety of rapidly expanding rural populations in Nepal and Bihar, India. This new approach adds a new tool in the determination of sediment flux and its role in changing flood risk linked to climate and land-use change.

**Acknowledgments**

We thank David de Klerk for the great coding support. We thank Yu Morishita for technical support during the SAR data processing. We thank Simon Mudd for support during the LSDtopo tool river network extraction processing. We thank Prakash Pokhrel for the discussion on the river catchment erosion rates and for implementing the LSDTopo tool for calculating river catchments. We thank Yasser Maghsoudi Mehrani from the Centre for the Observation and Modelling of Earthquakes, Volcanoes and Tectonics (COMET) for processing 20 m resolution Sentienl-1 interferograms and long timespan 100 m resolution Sentienl-1 interferograms. Last but not least, we want to extend our deepest thanks to Bill Hauer from the Alaska Satellite Facility (ASF) for his continued support with their toolbox and data. JH is supported by a Daphne Jackson Fellowship, funded by Natural Environment Research Council (NERC).

**Code availability**

The code used in this study for generating coherence time-series in Figure 4b, as well as the coherence data from Lazecký et al. (2020) used in the plot, is openly available and can be accessed here: https://zenodo.org/doi/10.5281/zenodo.13222093

**Author contribution**

JH and HDS conceptualized the study. JH developed the method, conducted the analysis, and drafted the manuscript. HDS reviewed and edited the manuscript.

**Competing interests**

The authors declare that they have no conflict of interest.



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

Zheng, Y. and Fattahi, H.: Modeling soil moisture with cumulated closure phase of interferometric SAR measurements, AGU 750 Fall Meeting Abstracts, H32K-05,