# Peer review of "Sediment aggradation rates in Himalayan rivers revealed through InSAR differential residual topographic phase"

_EGUsphere, 2024_

## Author Comment (AC1)

**Authors' Response to Reviews of**

Sediment aggradation rates in Himalayan rivers revealed through InSAR's differential residual topographic phase

J. Huang, H. D. Sinclair

EGUsphere Preprint, https://doi.org/10.5194/egusphere-2024-2600
* * *
*RC: Reviewers' Comment*, AR: Authors' Response, □ Manuscript Text

AR: Dear Dr. Johannes Leinauer,

Thanks so much for your interest in our work and your valuable comments. We really appreciate the time you took to share your suggestions. Your insights have greatly helped improve our manuscript, and we're truly appreciate the time you devoted to this manuscript.

Kind regards,

Huang and Sinclair

*RC: This is the first time I am reviewing the manuscript entitled: "Sediment aggradation rates for Himalayan Rivers revealed through SAR remote sensing". The authors describe a method to use differential residuals of SAR data to detect mm-scale elevation changes in the range of 20 mm/yr in four seasonally dry rivers of Nepal. They claim to "demonstrate the feasibility of InSAR techniques in geomorphological monitoring". The technical and geomorphological aspects of this study are generally interesting and have potential to bring forward this research field.*

*As I am not a specialist in SAR analysis, I will focus on the structure and storyline of the paper and the geomorphological implications/ interpretations.*

*Goal of the paper*

*First, the general goal or main storyline of the paper is not clear to me. I see two possibilities:*

*The goal is to prove that the suggested methodological approach can detect sediment dynamics in the selected rivers, or*

*The measured and processed signals support a geomorphological process that can now be understood or described better.*

*However, possibility 1) would require some prove that the results of the suggested methodological approach are true or at least reproducible and consistent with other methods.*

*Possibility 2) would require a clear story/ concept, of which processes should be described and supported. Then, the description of the methods should not be the main focus of the paper but rather be described as a tool to solve the stated geomorphological problem and the observed processes must be discussed in detail.*

AR: Thank you for suggesting the two possible storylines for the paper. I prefer the first option, as the majority of the paper focuses on describing this new InSAR approach - DRTP. The results are **reproducible** since we are using open data and open code for signal processing. However, this open code only works with high-quality residual topographic phase data and does not include corrections for the topographic phase ambiguity.

Our immediate next focus is to develop our own open code for DRTP SBAS-InSAR approach, tailored to more complex river systems. All conventional SBAS-InSAR approach is primarily for the line-of-sight displacement phase inversion. While we are adapting it for the differential residual topographic phase processing, we have to develop our own open code with a focus on river sediment mapping. This includes incorporating a machine learning pixel selection method focused specifically the dry river pixels, among other enhancements.

Regarding whether this new InSAR approach is **true**, it builds on the work of Zhang et al. (2019), who inverted all phase components into displacement first and then removed the 'noise'. Fattahi and Amelung (2013) demonstrated that, in the displacement time-series domain, the linear model achieves zero RMSE for the estimated residual topographic phase. Additionally, the mathematical equation describing the residual topographic phase (Bombrun et al., 2009) supports our methodology.

Our new approach is grounded in these previously published foundations, with the key innovation being the modification of the changing in the topographic height in Equation (7) in our paper. It is worth noting that an essential aspect of this approach, as with all other InSAR data processing, is the quality of the input data (high interferogram quality with strong residual topographic phase).

In terms of **consistency** with other methods, since this is the first observation detecting millimeter-scale riverbed elevation changes, there are no existing methods to compare with. One approach is to use the subsidence rate in the adjacent cropland, mapped using the conventional InSAR method based on the line-of-sight phase. Whether the conventional InSAR or the residual topographic phase InSAR from our new approach is used, as long as the input phase data is of a high quality, the results should indicate the same rate. For instance, in Figure 17, point C, which is adjacent to river 3, shows a subsidence rate of -14 mm/year based on conventional InSAR deformation phase. In Figure 14, river 3 near point C indicates a subsidence rate of -12 mm/year, with an uncertainty range of -12% to +8%, corresponding to rates between -13.4 mm/year and -11 mm/year based on the differential residual topographic phase.

***The conclusions state that this manuscript develops a "novel approach", provides "detailed, high-resolution geomorphological data", shows a "significant sediment aggradation" (is it statistically significant?) and that "this approach adds a new tool". These statements should be supported clearly be the main part of the manuscript.***

AR: Thank you for highlighting the need to emphasise the key points. In the Introduction, we have now clarified the novel aspects of using the DRTP approach, and highlighted this by a change in the manuscript title to 'Sediment aggradation rates in Himalayan rivers revealed through InSAR's differential residual topographic phase'. The DRTP approach introduces a novel method by treating the residual topographic phase as a signal rather than noise. Its phase difference is used to map sediment height changes, as described by Equation (7) in our paper. This new equation is a modified from Bombrun et al. (2009) and Fattahi and Amelung (2013). The DRTP approach allows for the tracking of elevation changes even in cases of land-cover change, where coherence is lost, making it

impossible to retrieve the line-of-sight displacement phase. These points are supported in lines 300–311 in the manuscript.

AR: The phase velocity standard deviation mentioned in Section 3.6 demonstrated the statistically significant of the result, which is less than 1 mm/yr of standard deviation. The velocity standard deviation is calculated based on a bootstrapping approach, which uses the cumulative displacement data and repeated bootstrap sampling from original cumulative displacement data, then calculates the velocity. The standard deviation of the velocity tells us how much the estimate velocity estimate varies. The 1 mm/yr of standard deviation is low, which means the displacement time-series is not noisy.

AR:

> 300    In our study, we use the SBAS-InSAR technique to invert differential residual topographic phase to elevation change rates. The inversion of the DRTP network for the estimated phase history is implemented in LiCSBAS software (Morishita et al., 2020) using the NSBAS technique, which assumes a linear deformation model. The phase history's effect is predominantly influenced by DRTP along the river channels, as expressed in Eq. (7). The variations of baseline resulting in small jumps within the same year during the dry season is caused by the topographic phase ambiguity (red-coloured component in Eq. (7)).
> 305    We maintained the perpendicular baseline within a range of ±100 meters because the majority of the interferograms fall within this range. The offset between the different years residual topographic phase includes the combination of topographic phase ambiguity and phase changes linked to changes in the height of the sediment. It is important to note that the differential topographic phase caused by river sediment aggradation is larger than the variations caused by the topographic phase ambiguity. The final elevation change rates are calculated from the residual topographic phase history based on Eq. (7). The
> 310    DRTP approach enables tracking of elevation changes even in cases of land-cover change, where coherence is lost, preventing the retrieval of the line-of-sight displacement phase.

*Structure*

*In general, the readability of manuscript could benefit from a clearer structure. There are two method sections (methodological background and methods applied in this study). This causes repetitions. The methods should be focused only on things that are needed to solve the focus problem of the study. Starting with the general principal could help non-SAR-specialists to follow. Additionally, some results and interpretations appear in the methods section (which polarization amplitude is higher, the effect of soil moisture, sources of errors and uncertainties…). Vice versa, in the results section, some things are shown that have been presented in the methods before.*

AR: Thank you for noting the two methods sections. We have updated the title of Section 3 to 'Methodology for DRTP InSAR application to dry gravel riverbeds.' We have also shortened the section 3.1, and moved most of the text into the supplementary material. After restructuring the manuscript, we have ensured that there is no repetition of text between the methods and results sections, and have given it a simpler structure that hopefully is clearer to the reader.

*The introduction and methods sections take 19 pages, results 3 pages, discussion 4 pages and conclusions 0.5 pages. However, the structure of the manuscript should somehow fit the scope of the paper. It might be possible to increase conciseness by re-evaluating, if all 16 figures are necessary to support the main goal.*

*The discussion about uncertainties appears as an own section before the section "Discussion". The section 8.1 "Validation…" consists of one paragraph giving an outlook on further research and only a brief comparison to other sediment aggradation rates. This could be elaborated in more detail.*

*Geomorphological processes*

*The positive elevation change is highest close to the mountain range. How can you exclude influences of topographic uplift of the mountains also raising the riverbeds? Fig. 17 touches this aspect by comparing the riverbeds to the surrounding areas. How can you make sure that the differences in elevation change are not influenced by the different datasets and processings (20 m vs. 100 m resolution)?*

AR: Thank you for highlighting the structural issues in the manuscript. We have restructured it, as illustrated in the figure below.

AR: The reviewer raises a good point here that we are keen to emphasise. Whilst there are localised areas in the Gangetic Plains where blind thrusts may cause structurally driven surface uplift, we can remove this as a signal in this area by demonstrating that the floodplains are subsiding. So the contrast with increasing elevation occurring solely in the channels implies that this process is linked to surface processes of sediment transport in the channels themselves.

AR: The 20 m and 100 m resolution datasets are produced from the same InSAR data, differing only in resolution. The variation in resolution is due to coherence levels: the riverbeds have high enough coherence to achieve a good signal-to-noise ratio at 20 m resolution. Both the 20 m and 100 m resolution results are processed using the same reference point. The differences in elevation change are likely governed by the phase information at the different land-cover (riverbeds and floodplain). At the riverbeds, the phase primarily has residual topographic phase contribution, while at the floodplain, it has both line-of-sight displacement phase and residual topographic phase contributions.

AR:

*Following your analysis, the surroundings close to the mountain margin lowered 60-90 mm over the studied time period and the river channels raised 70 mm. This means that the channel raised 130-160 mm relative to its surrounding. Is there a way to verify this?*

AR: For analysing elevation rate changes along riverbeds due to subsidence and sedimentation, we will use river 1 as an example (refer to the figure below). The solid red line represents the projected trend in sedimentation rates for River 1. The dashed red vertical line, showing a rate of approximately 10 mm/yr of the difference between the projected sedimentation rates and the observed elevation change rates at this pixel in river 1. The 10 mm/yr is from the subsidence effect. This indicates that the riverbed at the mountain front pixel raised by 12 mm/yr, while the adjacent floodplain subsided by 10 mm/yr. Both measurements reference to the same point, which is the airport located at the embanked, inactive riverbed.

AR: In addition, we have referred to the cross-validation between the channel and floodplain in our earlier response. We have also discussed more general validation of the results in the response to reviewer 3, and in the discussion section 6.2, lines 488-503.

AR:

[Figure]

*You interpret a "channel avulsion every few hundreds of years". If this is true, this should be possible to see in the geological record, detectable by geophysics or in outcrop profiles, and it might even be possible to date avulsion layers. Without further proof or discussion, this hypothesis stands alone.*

AR: We have added three additional references that discuss rapid avulsion frequencies in rivers immediately east (the Kosi) and west (the Bagmati) of our study site in lines 536-540.

AR:

[Figure]

*If the riverbed is incised into the surrounding floodplain, there must be an erosive process. How and when does erosion happen? If I understood right, then during monsoon there is aggradation and during the dry season there is no sediment change. If finally, the channel is filled up and avulsion happens, how does the channel erode into the surroundings again?*

AR: We are not convinced that there is any evidence of significant incision of the channel into the surrounding floodplain. Our Figure S4 in the supplementary material demonstrates that all of the channels are aggrading, with a few localised patches of net subsidence that may be caused by localised erosion of the channel. Once the channel is abandoned through avulsion, there is no further process to drive erosion.

---

## Author Comment (AC2)

**Authors' Response to Reviews of**

Sediment aggradation rates in Himalayan rivers revealed through InSAR's differential residual topographic phase

J. Huang, H. D. Sinclair

EGUsphere Preprint, https://doi.org/10.5194/egusphere-2024-2600
* * *
*RC: Reviewers' Comment*, AR: Authors' Response, □ Manuscript Text

AR: Dear Prof. Bookhagen,

Thanks so much for your interest in our work and your valuable comments. We really appreciate the time you took to share your suggestions. Your insights have greatly helped improve our manuscript, and we're truly appreciate the time you devoted to this manuscript.

Kind regards,

Huang and Sinclair

*RC: This manuscript describes an interesting and creative approach to measure sediment-height changes using radar-interferometric time series analysis. The author attempt to exploit the high temporal resolution of SAR data to better understand sediment dynamics. The authors rely on topographic residuals (or sometimes called DEM errors) and their changes through time to measure small height changes of sediment deposited in large rivers. This is an interesting approach, because standard radar interferometry will not allow to track height changes due to land-cover changes. This appears to be the first application of topographic residual analysis to sediment-transport studies. While this is creative, it is also tricky and has many caveats (see below). The authors partly field validate their measurements with general budgets, but not with measurements at the timescale of the SAR data and the presented signals have no uncertainties.*

AR: Thank you for summarizing the study and highlighting that this is the first application of differential residual topographic phase in the analysis of sediment-transport and accumulation. Since this study provides the first millimetre-scale measurements of sediment height changes covering 15 km stretches across four ephemeral Himalayan mountain-front rivers, no other measurements are available at this timescale (2016–2021) for cross-validation. The impact of variable perpendicular baselines on residual topographic phase is modelled using a linear height change model in the Supplementary Material 2 and found that the topographic phase ambiguity induced uncertainty ranges between -12% and +8%. As mentioned in lines 325–345 and shown by the shaded areas in Figure 14 (the uncertainties in range -12% and +8%).

AR:

325 In our study, we use the SBAS-InSAR technique to calculate the annual residual topographic phase. First the annual residual topographic phase is calculated, with variation of baseline caused small jumps within the same year during the dry season. In our study, we maintained the perpendicular baseline within a range of ±100 meters because the majority of the interferograms fall within this range. Then calculate the offset between the different years residual topographic phase. It is important to note that the differential topographic phase caused by river sediment aggradation is much greater than the small variations caused

330 by the variation of the perpendicular baseline. The five years (2017 − 2021) residual topographic phase with 4 differential topographic phase calculated. Finally those four differential topographic phase value converted into elevation change along dry gravel riverbeds. The differential residual topographic phase values are then converted to annual elevation change rates using Eq. (15). This procedure is implemented using the LiCSBAS code (Morishita et al., 2020).

To summarize, the SBAS-InSAR processing based on differential residual topographic phase relies on several assumptions:

335 (1) The residual topographic phase is the predominant phase value along the dry riverbeds, unaffected by noise and line-of-sight (LOS) displacement phase. To support this assumption, the background LOS displacement signal must be analyzed and separated. The time-series mapping of the basin background indicates that the LOS displacement remains 'flat' during the dry season (Fig. 18). Therefore, we assume that the phase observed along the dry gravel riverbeds is primarily from the residual topographic phase. Additionally, we examined the unwrapped phase profile along the river and its sensitivity to the

340 perpendicular baseline, which demonstrates a positive linear relationship between topographic phase sensitivity and the perpendicular baseline (Fig. 10); (2) The network connectivity of each acquisition time results in similar topographic phase sensitivity (phase ambiguity), as indicated by relatively flat time-series within the same year (Fig. 16); (3) We account for variations in the scaling factor by calculating the average perpendicular baselines for the five different connected networks are 52.2 m, 52.8 m, 49.5 m, 48.4 m, and 50.4 m (Fig. 8). Consequently, the ratios of $B_{12}/B_{11}$ are 1.01, 0.94, 0.94, and 1.04. To

345 quantify the uncertainty percentage caused by these ratios, we conducted forward modelling (Supplements 2) and observed their effect on the elevation change ratios to be +2%, -12%, -12%, and +8%. Therefore, we conclude that the impact of the scaling factor on the final result's uncertainty percentage falls within the range of +8% to -12% (Fig. 14).

*RC: The study focuses on the foreland of the Himalaya in the Ganges plains that show a strong sediment-flux dynamics. While there are several creative and interesting thoughts in the manuscript, there exist several points that need structuring and clarification. In the following, I am listing several points that should be looked at and considered during a revision process: 1) Methodological description. The method section starts out by explaining scattering and polarization and then explains amplitude measurements (the description of amplitude is after the scattering section – this should be reversed). This is followed by some backscattering analysis of rivers using pre-processed GRD data obtained from Google Earth Engine. While this is an interesting exercise (including Figure 3), it is irrelevant to the topographic residual (or DEM error) used for height-change mapping. These topics (several pages of text and figures) is also not picked up on in the Result, Discussion, and only somewhat in the Conclusion section.*

AR: Thank you for asking about the scattering and polarization. The rationale for describing scattering and polarization first is to clarify why gravel beds exhibit relatively stronger VV amplitudes compared to vegetated gravel bars. Conversely, VH amplitudes show relatively higher values compared to gravel beds, as demonstrated in Figure 2. This distinction underscores the potential of PolSAR amplitude time-series for detecting changes in riverbeds related to water level fluctuations and vegetation growth. We have added these points to the end of section 3.1 in lines 114-119. We have also shorten the section 3.1, moved the most text into the Supplementary Material 1.

might be less significant. Because the SBAS-InSAR method relies on distributed backscatter, which is particularly effective in areas with diffuse scattering of stronger VV polarization, it is important to explain upfront the backscattering type and polarization characteristics of the dry gravel riverbeds. Additionally, applying this novel DRTP approach to more complex rivers, beyond ephemeral rivers, requires classifying dry gravel pixels based on SAR amplitude polarization characteristics and their statistical metrics. It is important to use SAR amplitude for classification instead of optical or multi-spectral images, as the same SAR images' phase component is used in the DRTP approach to map sediment aggradation rates.

AR:

**RC: The section on SAR coherence is important, but it needs to be clarified what coherence is shown – averaged spatial coherence or temporal coherence (as it mostly used because it accounts for the temporal decay of coherence or decorrelation – see Figure 4).**

AR: Figure 4 shows both averaged spatial coherence and temporal coherence. The figure caption for Figure 4 is now revised in line 220-225 in the revised manuscript.

Figure 4: (a) The averaged spatial coherence map with 20 m resolution across the study area during the season; (b) Temporal coherence time-series at the black point in (a) on river 2 shows seasonal variation. Within the same year, the short timespan coherence is higher during the dry season and lower during the monsoon season, probably due to the waves on the river surface causing low coherence. The long timespan interferograms that cross two dry seasons exhibit low coherence, likely due to sediment erosion and deposition during the migration of channels and bar-forms caused by the monsoon floods; (c) Spatial coherence value were plotted along river channels providing insights into the spatial variability with troughs at 0.3 and peaks at 0.8. The coherence troughs are typically found at the edges of the channel and vegetated sand bars, which results in fewer data points in the final InSAR results. These areas of low coherence do not show a noticeable alteration in the trend of InSAR elevation change results, but the data points in the trend are more scattered (Fig. 14). © Google Earth

AR:

**RC: There are two method section – this is awkward. Here is space to consolidate and significantly shorten the manuscript (to make room for more important analysis – see below).**

AR: Thank you for noting the two methods sections. We have updated the title of Section 3 to 'Methodology for DRTP InSAR application to dry gravel riverbeds.' We have shorten the section 3.1, moved the most text into the Supplementary Material 1.

**3 SAR backscattering on dry gravel riverbeds**

**3.1 SAR polarimetric backscatter amplitude analysis for dry gravel riverbeds**

The backscatter energy (amplitude) is a critical parameter in SAR data analysis as it provides information about the surface scattering properties and is used to derive surface characteristics. Usually, higher backscatter energy results in clearer and more

AR:

**RC: Coherence Thresholds: I am not certain where the authors picked their coherence thresholds from, but these are not typical (they are too low). The statement that coherence above 0.3 is useful cites a study Cigna and Sowter, 2017) that uses ISBAS (a different method) and this is not relevant for SBAS (or NSBAS).**

AR: ISBAS, NSBAS, and other members from the SBAS family are all modified from the conventional SBAS approach were developed in the early 2000s by Berardino et al. (2002). The conventional SBAS method applies least-squares to solve the velocity inversion, converting phase into displacement. Since 2002, numerous variations of SBAS, including ISBAS and NSBAS, have emerged and been published. While all SBAS methods share fundamental principles, each employs slightly different techniques to address issues like network gaps and low coherence. For

example, LiCSBAS follow NSBAS technique to addresses disconnected networks using a 'temporal constraint,' which adds linear movement between the gaps. This approach resolves the network gaps issue during the inversion. Meanwhile the ISBAS techniques is more 'relaxed' in selecting pixels that are coherent, instead of whole stack of the interferograms with high coherence, it allow only part of the interferogram stack with high coherence (also called intermittently coherent). Accordingly, only the partial high coherence pixel's phase value used in the inversion. In our study, a coherence threshold of 0.3 was used to exclude long timespan interferograms. For short timespans, the average coherence value along the dry gravel riverbeds is 0.6, as shown in Figure 4(c).

*RC: SAR-Data characteristics. Throughout the manuscript, the authors refer to 20m and 100m data. This is very unusual. You usually give the number of multilooks (range/azimuth), because this better reflects data characteristics. There is certainly an equivalent in square area, but the multilook values are more common and it is also not exactly 20 or 100 m.*

AR: Thank you for the opportunity to explain the concepts of range, azimuth, and multi-look. Multi-look numbers of range and azimuth better represent the original SAR image acquisition projection, where azimuth aligns with the satellite's flight path, and range is perpendicular to it. For Sentinel-1 SAR satellite, range resolution is approximately 5 m, and azimuth is around 20 m. A low multi-look setting (range 4, azimuth 1) performs well in high-coherence areas, with an average coherence 0.6 of dry gravel riverbeds, supporting 20 m resolution for mapping riverbeds. Since the riverbeds in our study area is about 300 meters wide, maximizing pixel count is a priority. The corresponding text is now updated in revised manuscript line 213.

AR:

> 210  resolution (along-track direction) is approximately 20 m. The low multi-look value of range 4 and azimuth 1 only works at the high coherence area, which in this case is the dry season gravel riverbeds (Fig. 4). Again, 20 m resolution SAR images are solely used to map elevation change along dry riverbeds. For the rest of the study, we will quote azimuth and range in terms of pixel size, which corresponds to an approximate resolution of 20 meters per pixel.

*RC: I am puzzled by the statements about SBAS. They argue they have used SBAS, but the method implemented by Morishita in LiCSBAS is NSBAS that can also deal with disconnected networks (but require additional terms). The networks shown in Figure 8 and 9 are all disconnected. You can not use SBAS to work with them in a reliable manner. Connected networks are always more reliable than disconnected networks during an inversion. In Figure 9 there is the note that the network has been linked though linear fitting. Is this not NSBAS? The section on the network inversion needs more work (and also more information for the reader to see number of ifgs, images, baselines and coherence history). 2. General information on data, processing, and methodology. The authors never tell us what scenes (track/frame/bursts/swath) and how many connections have been used. What is the number of interferograms? It is also not clear what software has been used and what parameters. I assume LiCSBAS was used, but no information is given on the SAR processing and interferogram generation.*

AR: LiCSBAS follow NSBAS technique to addresses disconnected networks by fitting an overall linear trend to the time-series (Morishita et al., 2020). Mathematically, LiCSBAS uses singular value decomposition to solve the inversion when gaps exist in the network, with a modification called 'temporal constraint,' which adds linear movement between the gaps. This approach resolves the

network gaps issue during the inversion. While the 'linear fit' functionality is useful, it is important to assess the composition of the phase value and the basic trend of elevation change to determine whether this 'linear fit' is suitable.

In our study, the primary elevation change in the floodplain is driven by subsidence, likely caused by groundwater extraction and seasonal recharge. When small gaps of a few months occur, the LiCSBAS approach is applicable. We also attempted to fill as many gaps as possible by using long timespan interferograms (Fig. 9).

For the dry gravel riverbeds, the land-cover changes completely each year as new sediment is deposited on top of the old sediment. This means a complete loss of coherence, making it impossible to generate an interferogram to fill the network gap. The elevation change is dominated by the height increase from this new sediment layer, so the phase value in the dry gravel riverbed is primarily influenced by residual topographic phase. In this case, along the riverbeds, the gaps are filled by its own differential residual topographic phase value, as illustrated in Figure 10 (the phase value difference between the blue and black curves). To clarify this, we have removed the text in Figure 9 caption stating 'the network has been linked through linear fitting'.

We have added a table in the Supplementary Material 1 listing the SAR images used for mapping height changes in dry gravel riverbeds. The SAR images are sourced from the Alaska Satellite Facility (ASF) and processed into InSAR images by the Centre for the Observation and Modelling of Earthquakes, Volcanoes, and Tectonics (COMET), as mentioned in the acknowledgments.

| Sentinel-1 Descending SAR Images (YYYYMMDD), path 121/frame 504/Interferometric Wide Swath mode | | | | | | | | | | | |
|---|---|---|---|---|---|---|---|---|---|---|---|
| Date | Baseline (m) | Date | Baseline (m) | Date | Baseline (m) | Date | Baseline (m) | Date | Baseline (m) | Date | Baseline (m) |
| 20161027 | 46.7 | 20170113 | 111.6 | 20180102 | 23.5 | 20190109 | 30.4 | 20200104 | -13.7 | 20210110 | -14.3 |
| 20161102 | 77.8 | 20170119 | -62.0 | 20180114 | 43.6 | 20190121 | 93.6 | 20200110 | 74.9 | 20210122 | -48.4 |
| 20161108 | 39.0 | 20170206 | -19.8 | 20180126 | -30.4 | 20190202 | 81.1 | 20200116 | -6.0 | 20210203 | 12.0 |
| 20161126 | 3.8 | 20170212 | 103.9 | 20180207 | 21.0 | 20190214 | -34.5 | 20200122 | 37.0 | 20210215 | 72.3 |
| 20161202 | 87.2 | 20170308 | -60.1 | 20180303 | 61.2 | 20190226 | 26.6 | 20200128 | 48.2 | 20210227 | 60.6 |
| 20161214 | 74.0 | 20170320 | -56.7 | 20180315 | 17.7 | 20190310 | 75.6 | 20200209 | 104.9 | 20210311 | 27.3 |
| 20161220 | 63.9 | 20170401 | -8.7 | 20180327 | -53.8 | 20190322 | 85.9 | 20200221 | 34.7 | 20210323 | -64.7 |
| 20161226 | 54.1 | 20170413 | 40.4 | 20180408 | -74.0 | 20190403 | 81.5 | 20200304 | -17.7 | 20210404 | -86.5 |
| | | 20170425 | 46.0 | 20180420 | -89.2 | 20191012 | -16.8 | 20200316 | 0.5 | 20210416 | -35.3 |
| | | 20171103 | 63.5 | 20181017 | 41.6 | 20191018 | 40.4 | 20200328 | 22.5 | 20210428 | -23.3 |
| | | 20171115 | 14.4 | 20181029 | 46.9 | 20191024 | -83.8 | 20200409 | 53.1 | | |
| | | 20171127 | 11.0 | 20181110 | 66.4 | 20191030 | -15.0 | 20201111 | 21.3 | | |
| | | 20171209 | -40.3 | 20181122 | 33.1 | 20191105 | 36.6 | 20201123 | -19.9 | | |
| | | 20171221 | 55.6 | 20181204 | 53.1 | 20191111 | 29.7 | 20201205 | 60.2 | | |
| | | | | 20181216 | -62.9 | 20191117 | 75.6 | 20201217 | 112.0 | | |
| | | | | 20181228 | 54.5 | 20191123 | 38.9 | 20201229 | 72.1 | | |
| | | | | | | 20191129 | 89.9 | | | | |
| | | | | | | 20191205 | -6.2 | | | | |
| | | | | | | 20191211 | 45.2 | | | | |
| | | | | | | 20191217 | 66.0 | | | | |
| | | | | | | 20191223 | 29.4 | | | | |
| | | | | | | 20191229 | 102.6 | | | | |

Table S1: List of SAR images and their associated baselines used for river sediment aggradation mapping. The perpendicular baselines are relative to the reference SAR image dated 2016-09-21.

AR:

*RC: 3. Topographic Residuals or DEM errors. This is the core, creative part of the study. The authors should carefully introduce the topographic residuals and their caveats. The authors are also not the first ones using topographic residual for deformation measurements (but likely the first one to apply it to sediment-height dynamics). I remember there was a study to use topographic*

*residual from ALOS data to measure lava thickness: Measuring large topographic change with InSAR: Lava thicknesses, extrusion rate and subsidence rate at Santiaguito volcano, Guatemala, https://www.sciencedirect.com/science/article/pii/S0012821X1200194X They do something similar but using the ALOS-L band and they make sure to use large baselines (see below) and use synthetic models to get a better understanding of uncertainties. The study by Bombrun et al., 2009 (10.1109/LGRS.2009.2026434) is also something to look at. Importantly, topographic residual is a tricky beast. It is a relative error. It is relative to the network and relative to space. That is, changing the network structure or moving the reference point will result in different topographic residuals.*

AR: We have now added in line 325 to introduce the topographic residuals and their caveats, emphasizing that the differential topographic phase caused by river sediment aggradation is greater than the small differences caused by variations in the perpendicular baseline.

AR: Ebmeier et al. (2012) estimated the height difference between newly deposited volcano lava flows ($\geqslant$25 m thick) and the DEM based on the **absolute** residual topographic phase with an average uncertainty in lava thickness of ~±9 m, based on four pair L-band interferograms's **absolute** residual topographic phase. In contrast, we measure elevation change by calculating the **differential** residual topographic phase with millimetre-scale accuracy over the river sediment aggradation rates. Accuracy is improved by using a stack of interferograms, with the difference between each year's residual topographic phase being calculated and then inverted into millimeter-scale elevation change rates.

We want to emphasize that regardless of whether **absolute** residual topography or **differential** residual topography is used, both are tools for mapping topographic height changes due to sediment accumulation, not for measuring deformation. Deformation measurement is defined as tracking elevation changes when land cover remains coherent, typically due to internal factors such as aquifer compaction or fault movement. With coherent land cover, the line-of-sight deformation phases are measured, then inverted it into line-of-sight displacement rates. However, sediment-related height changes naturally alter land cover, resulting in loss of coherence completely. In such cases, deformation phase measurement is impractical, as it relies on high coherence. Therefore, residual topographic phase offers an alternative to measure the elevation change in low-coherence situations.

To leverage the residual topographic phase, first is to accurately retrieve the topographic phase. There are two methods to increase the accuracy of the retrieved topographic phase, **multi-temporal method and time-series domain method.** Bombrun et al. (2009) detailed the mathematical framework and methodology for using differential residual topographic phase in InSAR to estimate residual DEM. They focused particularly on height ambiguity, which is predominantly controlled by the perpendicular baseline. Du et al. (2016) did a detailed study of comparing different multi-temporal methods (PS-InSAR, LS solution SBAS, SVD solution SBAS, and SVD combine with LS solution SBAS) with the impact of four factor (baseline thresholds, interferogram quality, network connectivity, deformation assumption), how accurate they could retrieve residual topographic phase into residual DEM. Du et al. (2016) demonstrated that an SVD-based SBAS solution with a linear model, has low sensitivity to baseline threshold but is highly impacted by interferogram quality, network connectivity, and deformation assumptions. Bombrun and Du deal with the residual topographic phase in interferogram domain, so the RMSE of estimated residual DEM value is in meter-level.

In 2013, Fattahi and Armleng were the first to demonstrate the multi-temporal differential residual topographic phase in the time-series domain. This shift from the **interferogram domain** to the **time-series domain** significantly improved the accuracy of residual topographic phase correction operation. Fattahi and Amelung (2013) demonstrated the effect of estimate DEM error on displacement time-series with zero RMSE in linear deformation history. Their study also showed the effectiveness of using phase velocity history in time-series domain instead of interferogram domain. However, previous studies primarily treated the residual topographic phase as noise, aiming to retrieve and remove it from the displacements time-series results to preserve the integrity of the line-of-sight displacement trends and rates. In our study, we leverage the differential residual topographic phase in time-series domain for the cases that involve topographic height change. The residual topographic phase varies due to height changes caused by sediment dynamics, and then we retrieve the millimetre-scale accuracy height change based on our novel **Differential Residuals Topographic Phase (DRTP) technique**.

While millimetre-scale accuracy in height change mapping can be achieved using differential residual topographic phase through multi-temporal method in time-series domain, certain limitations of this approach should be noted. Since the RMSE is zero for linear deformation histories, this method is suitable for mapping sediment dynamics where the background deformation trends are linear. The primary source of uncertainty in mapping riverbeds arises from the scaling factor caused by topographic phase ambiguity in Equation (7).

AR:

> 325 In our study, we use the SBAS-InSAR technique to calculate the annual residual topographic phase. First the annual residual topographic phase is calculated, with variation of baseline caused small jumps within the same year during the dry season. In our study, we maintained the perpendicular baseline within a range of ±100 meters because the majority of the interferograms fall within this range. Then calculate the offset between the different years residual topographic phase. It is important to note that the differential topographic phase caused by river sediment aggradation is much greater than the small variations caused
> 330 by the variation of the perpendicular baseline. The five years (2017 − 2021) residual topographic phase with 4 differential topographic phase calculated. Finally those four differential topographic phase value converted into elevation change along dry gravel riverbeds. The differential residual topographic phase values are then converted to annual elevation change rates using Eq. (15). This procedure is implemented using the LiCSBAS code (Morishita et al., 2020).

*RC: Reference Point: I am surprised to see the reference point to be far away from the actual stream studied. The coherence is low and it looks like there are disconnected components – which is a problem for unwrapping. How were the disconnected components connected?*

AR: The airport was chosen as the reference point because it is located on the non-active riverbed with high coherence, indicated by the red-coloured high coherence values in Figure 6. Additionally, the airport's buildings are stable. The reference point must be located within the processed InSAR area and should not be placed on the active river channel, as the channel is the target for elevation change monitoring, with changes occurring annually. Therefore, the reference point should be within the processed InSAR area and exhibit minimal elevation change compared to other locations in the InSAR processed area. While I agree that subsidence may be occurring across the entire Terai region, including the non-active riverbed, the high heterogeneity of subsidence in the Terai region needs to be mapped. In our study, as seen in Figure 6(b), the unfiltered wrapped phase value along the riverbed remains within (π, -π), suggesting a low phase gradient that supports an accurate phase unwrapping result. Additionally, the multi-temporal InSAR approach effectively mitigates unwrapping errors. So far, preliminary results (see Figure below) and nearby GPS rate at a town called Biratnagar suggest that the subsidence rate at the airport is negligible, making the influence of the reference point minimal. The preliminary result,

derived from Sentinel-1 at a 100-meter resolution, demonstrates a reliable trend in the rates. However, significant noise contamination, primarily caused by vegetation in cropland, leads to a high spatial variation in the result, as shown in Figure 15 in the manuscript.

AR:

[Figure]

AR: http://geodesy.unr.edu/NGLStationPages/stations/BRN2.sta

[Figure]

*RC: The authors cite Du et al. 2016 and this is a detailed study of topographic residual measurements. Du et al. point to several of the above problems, especially the network structure. In order to estimate the impact of the network (and individual connections), one can randomly (?) remove connections to observe how the topographic residual changes. This is also a useful way to estimate uncertainty of the topographic residual. I am not fully clear how the networks shown in Figure 8 were connected, but the topographic residual is likely to be different between these years just because of a change in network structure (the magnitude of this signal can be*

*identified or modeled). Most importantly, I am puzzled by the approach to keep the smallest baselines. Topographic residuals are larger for large baselines – in other words, baselines exerts a significant sensitivity on topographic residuals. For interferometric measurements you are aiming at very small baselines to minimize the topographic effect (standard InSAR). I ask the authors to think about the signal they are looking for (if I understand them correctly): Larger baselines would be more appropriate for measuring topographic residual, because the measured signal is larger (see Figure 4 in Du et al.). If you confine the perpendicular-baseline tube to a very narrow range, you are optimizing the network for interferometric purposes, not for topographic residual measurements. This may sound counterintuitive and I may be missing parts of the author's explanation, but to enhance the topographic residual signal you are aiming at long perpendicular baselines, because these are more sensitive to topographic changes (see equation 15). One important issue only briefly addressed in the manuscript is atmospheric phase screening or tropospheric delay. The author's mentioned that they have used GACOS to correct their data, but no magnitude of the correction is shown and no dynamics of the tropospheric signal. This is important, because the tropospheric delay signal may easily exceed the topographic residual signal. Again, I urge the authors to look at Du et al. – they also have looked at different atmospheric delay patterns. The monsoon season in the Himalayan foothills is characterized by heavy, localized rainfall that may generate extreme delay signals. These turbulent components are not corrected for with ERA5 or GACOS data. However, the general water vapour content is captured. A simple question to ask: Is the topographic residual signal the same without any atmospheric correction (or a different correction)? A more careful treatment of the atmospheric correction and their impact is important, because this signal may have the same amplitude. This is also what Fattahi and Amelung (2013) stated in DEM Error Correction in InSAR Time Series; Heresh Fattahi and Falk Amelung, 2013 https://ieeexplore.ieee.org/abstract/document/6423275 (this is also cited).*

AR: In a detailed study of residual topographic phase measurement, Du et al. (2016) highlighted the impact of network connections on phase accuracy. One significant effect is from the perpendicular-baseline network. Du et al. (2016) demonstrated simulated data with a considerably larger range of perpendicular baselines, spanning from -200 to 400 meters. This large baseline range increases height sensitivity in residual topographic phase measurements. In contrast, Sentinel-1's precise orbital control keeps most SAR image acquisitions within a narrower baseline range of -100 to 100 meters, which has lower sensitivity in residual topographic phase measurements. Fattahi and Amelung (2013) demonstrated that a small baseline minimizes phase contributions from DEM errors; however, the multi-temporal approach will still cause small jumps in the residual topographic phase displacement history.

AR: To keep the effect of the residual topographic phase caused height ambiguity history minimal is important in our study, because we are leveraging the difference between the residual topographic phases to map the river sediment aggradation rates. Due to both the effect from the residual topographic phase ambiguity and the river sedimentation caused height increase, the residual topographic phase caused by the height increase must exceed the variation introduced by baseline-related displacement history.

Thank you for highlighting the use of residual topographic phase ambiguity in estimating uncertainty. In our study, we refer to Fattahi and Amelung (2013), who reported that the RMSE of differential residual topographic phase estimates is zero under a linear deformation history. The impact of variable perpendicular baselines on residual topographic phase is modelled using

a linear height-change model in the Supplementary Material 2 and found that the topographic phase ambiguity induced uncertainty ranges between -12% and +8%.

AR: The riverbed mapping interferograms were collected during the dry season only. We applied GACOS for atmospheric phase correction. In the dry season, atmospheric noise is characterized as cumulus clouds in fair-weather, making it easier to distinguish the atmospheric noise from the residual topographic phase trend along the riverbeds. Testing results with and without GACOS correction showed minimal difference, the figure now is added in the Supplementary Material 1 as Figure S5. The magnitude of the GACOS correction is approximately 1 radian, which are about 1.5 km dimple in size—typical for cumulus clouds in fair-weather conditions during the dry season in the Terai region. For monsoon season, however, atmospheric corrections should be carefully removed, as regional atmospheric effects may be stronger and cover the entire 15 km long river channels. Although single InSAR interferograms may be more impacted by atmospheric noise, a multi-temporal approach like SBAS-InSAR effectively minimizes these effects by analysing a large stack of interferograms in the time-series domain. Additionally, the dry season phase values are likely not unaffected by strong ionospheric phase delay. The rivers that are analysed in this study are geomorphological ideal for the DRTP application.

AR:

[Figure]

**Figure S5: Residual** topographic phase changes along river 2, with and without atmospheric correction. The filtered, unwrapped phase values with GACOS (brown curve), comparing without GACOS correction (blue curve). The GACOS correction revealed a couple of dimples in phase values near the mountain front, while the overall trend of the phase values remained the same. This indicates a minor atmospheric noise effect.

*RC: I mentioned it before, but I am surprised about the treatment of the coherence and connected components. Figure 16 shows the lower multilooking data (what the authors call 20 m resolution) and it looks like as if the individual rivers are not connected and have not been unwrapped together. The reference point appears to be outside the center stream of the plot. Is there a special treatment for connecting the components? The higher multilooking data (called 100 m resolution) appears to be connected but shows different signals. It is difficult to interpret these plots without additional information (also on the network). Maybe a coherence matrix through time would be helpful to better understand the interferometric network. I point out that other researcher have made careful statements before: "Noisy acquisitions with severe atmospheric delays or decorrelation noise could potentially bias the estimation of topographic residuals, the average velocity or coefficients of any temporal deformation model." (from "Small baseline InSAR time series analysis: Unwrapping error correction and noise reduction" by Zhang Yunjun, Heresh Fattahi, Falk Amelung , https://www.sciencedirect.com/science/article/pii/S0098300419304194*

AR: Interferogram network connectivity is solely due to its coherence, which vary based on the characteristic of the land-cover, independent from the multi-look resolution. There is no special method for connecting interferograms. Certain types of land cover, such as cropland, exhibit seasonal variations in coherence due to crop growth cycles. During the same growing season, longer time intervals tend to show higher coherence compared to shorter time intervals that span different growth periods. Conversely, riverbed areas experience complete coherence loss during monsoon seasons. This is caused by new sediment deposits, which preventing the interferogram computation across seasons.

Zhang et al. (2019) addresses unwrapping errors and their correction. In our study, as illustrated in Figure 6(b), the unfiltered wrapped phase values along riverbeds remain confined within the range of ($\pi$, -$\pi$), indicating a low phase gradient and supporting accurate phase unwrapping results. Furthermore, the multi-temporal approach effectively reduces unwrapping errors. LiCSBAS applies the closure phase method to filter out incorrectly unwrapped interferograms. Future work will investigate the phase gradient thresholds for the unwrapped phase in the DRTP approach.

***4. The Discussion starts well after 480 lines into the manuscript. It is very short and touches upon some relevant sediment-dynamics points. But none of the uncertainties of the topographic residuals or the tropospheric delays are discussed here. The section on future prospects is certainly important, but should not take up 1/3 of the Discussion section.***

AR: We added lines 557-591 in the discussion section in the revised manuscript.

AR:

> **6.6 Future applications of the DRTP methodology**
>
> A number of previous studies have considered methods to accurately retrieve the residual topographic phase in order to remove it. Bombrun et al. (2009) detailed the mathematical framework and methodology for using differential residual
> 560  topographic phase in InSAR to estimate residual DEM values. They focused particularly on height ambiguity, which is predominantly controlled by the perpendicular baseline. Fattahi and Amelung (2013) were the first to demonstrate the multi-temporal differential residual topographic phase in the time domain. This shift from the frequency-domain to the time-domain significantly improved the accuracy of residual topographic phase measurements. Notably, the Root Mean Square Error of the estimated DEM error is close to zero for both the linear and exponential displacement histories (Fattahi and Amelung, 2013).
> 565  Du et al. (2016) compares different multi-temporal approaches to reliably retrieve accurate topographic residuals in frequency domain. They demonstrated that a singular value decomposition based SBAS solution with a linear model, has low sensitivity to baseline threshold but is highly impacted by interferogram quality, network connectivity, and deformation assumptions. Ebmeier et al. (2012) estimated the height difference between newly deposited lava flows ($\geq$25 m thick) and the DEM based on the absolute residual topographic phase.

AR:

570   In this study, for the first time, we quantify elevation change caused by river aggradation and are able to map this with a millimetre scale accuracy by leveraging, the multi-temporal differential residual topographic phase displacement in time domain. In our case, the phase history is influenced not only by the perpendicular baseline history but also by the change in river sedimentation height. Consequently, we have updated the differential residual topographic phase mathematical formula in Eq. (7) to account for changes in height and introduced, for the first time, the concept of a scaling factor when using residual

575 topographic phase for mapping elevation changes. The priority for follow-on research is eliminating uncertainties caused by scaling factor effects in this novel approach (Zhang et al., 2019; Fattahi and Amelung, 2013). At the end, the success of this approach depends primarily on obtaining high-quality residual topographic phase data. High quality implies minimal noise contamination from phase unwrapping errors, atmospheric noise and other noises. For example, we applied GACOS for atmospheric phase correction. In the dry season, atmospheric noise is assumed to be randomly distributed clouds, making it

580 easier to distinguish the atmospheric noise from the residual topographic phase trend along the riverbeds. Testing results with and without GACOS correction showed minimal difference, the figure now is added in Appendix as Figure S5. The magnitude of the GACOS correction is approximately 1 radian within the dimples, which are about 1.5 km in size, typical for cumulus clouds in fair-weather conditions during the dry season in the Terai region.

   Additionally, PolSAR amplitude time-series could be used to detect changes in riverbeds from inundated to exposed,

585 demonstrating the potential for applying the DRTP approach to larger, non-ephemeral rivers globally. The DRTP SBAS-InSAR results in our study are influenced by the height ambiguity effect and elevation changes caused by sediment aggradation. A dedicated DRTP SBAS-InSAR software will be developed for river sedimentation rates mapping, eliminating the height ambiguity effect (Zhang et al., 2019; Fattahi and Amelung, 2013). Such research will help validate the robustness and scalability of this novel approach for its operational potential in developing its use as a standard tool in geomorphic and

590 hydrological research worldwide. Looking ahead SAR remote sensing will likely become standard practice for monitoring change in fluvial sedimentation rates globally.

*5. It took me a while to understand the vertical rates (I am still not certain that I understood the authors explanation). The Vertical rate are derived from the linear fit of the annual topographic residuals? Is the data in Figure 15 the slope of that linear regression? Are there uncertainties associated with that fit? 6. There are several useful figures in the manuscript. In general, it may be useful to convert the figures showing radians to mm, because the text argues about deposition (or sedimentation) rates in mm.*

AR: The term "vertical rates" refers to the rates of vertical elevation change. In this manuscript, vertical elevation change rates include two main components: vertical subsidence, likely due to aquifer compaction, and vertical sedimentation rates from river sediment aggradation. Thus, the vertical rates represent either of these two components or a combination of both. For example, in the Terai basin, the primary component is vertical subsidence rates. In downstream riverbeds, vertical sedimentation rates are the primary component, while in upstream riverbeds near the mountain front, both vertical subsidence and sedimentation rates play significant roles. Vertical subsidence rates indicate subsurface changes affecting elevation, whereas sedimentation rates represent surface processes influencing elevation. This distinction is crucial due to coherence effects, requiring the selection of the appropriate phase component to accurately map these elevation change rates.

Figure 15 shows InSAR-derived subsidence rate transects for the Terai basin. The rates exhibit high spatial variance due to significant noise contamination in the interferogram, primarily caused by vegetation in cropland. Despite this variance, the mean vertical subsidence rate across the basin is approximately -15 mm/year.

In Figures 16 and 17, which show time-series data for elevation changes in dry gravel riverbeds (Fig. 16) and the Terai basin (Fig. 17), the linear fit represents a linear interpolation applied to the displacement time-series. This is to address gaps in the interferogram network. As described in the manuscript (line 325-333), the phase used in the riverbed inversion is the differential residual topographic phase. Although gaps exist in the network, values from the differential residual

topographic phase allow for continuity in the time-series. The offsets observed in Figure 16 cross the gaps in the time-series based on differential residual topographic phase values.

AR: We have updated the Figure 10b with the unwrapped phase value converted to mm. Based on one phase equal to half wavelength is because the electromagnetic wave's two way travel between satellite and earth surface. The wavelength of the Sentinel-1 C-band SAR is approximately 5.5 cm, half wavelength is 2.7 cm, which is 27 mm.

***RC: There is no Figure 18, although it was mentioned several times.***

AR: Figure 18 in the text should be Figure 17, now is corrected in the text, thanks!

***RC: Overall, I see large potential in this study. It will require additional work if this is supposed to become a landmark study to propose topographic residual measurements for estimating sediment dynamics for current and future SAR missions (as suggested by the Discussion section). A thorough investigation of the impact of the interferometric network structure (including perpendicular baselines, temporal baslines, number of connection), tropospheric impact, and inversion approach will help to better understand boundary conditions and measurements uncertainties.***

AR: Thank you for recognizing the significant potential of the DRTP approach for sediment dynamics mapping. We have now updated the manuscript to include an investigation of the impact of the interferogram network (Figure S6), the tropospheric effects (Figure S5), and the inversion approach (lines 299-312).

AR: The NSBAS implemented in our study is using singular value decomposition to solve the phase velocity history inversion, with assumed linear model. The LiCSBAS implementation (Morishita et al., 2020) does not include invert the residual DEM. The MintPy implementation (Zhang et al., 2019) is using singular value decomposition and least squares to solve the phase velocity history inversion, with no assumed deformation model, and include the residual DEM calculation. Then the residual DEM is used to remove the residual topographic phase in time-series domain. In our study, the InSAR phase along riverbeds is assumed to be pure residual topographic phase, the LiCSBAS implementation inverted the differential residual topographic phase into the velocity. We interpret the velocity results based on Eq. (7), which include the effects from both height ambiguity and sedimentation caused height change. Our immediate next focus is to develop our own open code for DRTP SBAS-InSAR approach, tailored to more complex river systems.

AR:

[Figure]

Figure S6: (a) A network of 20 m resolution interferograms was added with 7 additional interferograms in 2016, each with an over 500 m baseline. (b) The red-coloured result from the interferogram network in (a). The black-coloured result represents baselines constrained within a ±100 m range, as illustrated in Figure 8. Both vertical rate curves exhibit the same trend: a 'V-shaped' subsidence between the mountain front and the forest edge, gradually decreasing towards gravel-sand transition, and finally fluctuating around zero. The vertical exaggeration in this plot is 250,000. The red curve is approximately 30% lower than the black curve due to the change in the topographic phase ambiguity caused by the 500 m baseline. Blue curve is modelled 30% less of the black curve.

AR:

In our study, we use the SBAS-InSAR technique to invert differential residual topographic phase to elevation change rates. The inversion of the DRTP network for the estimated phase history is implemented in LiCSBAS software (Morishita et al., 2020) using the NSBAS technique, which assumes a linear deformation model. The phase history's effect is predominantly influenced by DRTP along the river channels, as expressed in Eq. (7). The variations of baseline resulting in small jumps

305  within the same year during the dry season is caused by the topographic phase ambiguity (red-coloured component in Eq. (7)). We maintained the perpendicular baseline within a range of ±100 meters because the majority of the interferograms fall within this range. The offset between the different years residual topographic phase includes the combination of topographic phase ambiguity and sediment height change caused phase change. It is important to note that the differential topographic phase caused by river sediment aggradation is larger than the variations caused by the topographic phase ambiguity. The final

310  elevation change rates are calculated from the residual topographic phase history based on Eq. (7). The DRTP approach enables tracking of elevation changes even in cases of land-cover change, where coherence is lost, preventing the retrieval of the line-of-sight displacement phase.

---

## Author Comment (AC3)

**Authors' Response to Reviews of**

Sediment aggradation rates in Himalayan rivers revealed through InSAR's differential residual topographic phase

J. Huang, H. D. Sinclair

EGUsphere Preprint, https://doi.org/10.5194/egusphere-2024-2600
* * *
*RC: Reviewers' Comment*, AR: Authors' Response, □ Manuscript Text

AR: Dear Referee,

Thanks so much for your interest in our work and your valuable comments. We really appreciate the time you took to share your suggestions. Your insights have greatly helped improve our manuscript, and we're truly appreciate the time you devoted to this manuscript.

Kind regards,

Huang and Sinclair

*RC: I am not an expert in remote sensing or interferometry, so my focus will be on asking questions which I feel the text could benefit from addressing (either to help non-specialists such as myself to understand the paper, or to address the question directly).*

*The authors use interferometric techniques on repeat SAR observations to estimate sediment aggradation rates of gravel bedded rivers at the base of the Himalaya in the Ganges plains. They compare these rates with subsidence estimates from the local floodplains in order to show a very interesting, and novel methodology. They use these results to explore flooding and avulsion risk in this area, and discuss ways in which the method could be further validated.*

*Major Comments:*

*I enjoyed the paper, and the methods developed here are very creative and interesting. I found the quality of the science to be good. The presentation quality is the biggest issue for me, as the paper is confusingly structured, has some grammatical issues which make reading it unclear at times, and has calls to figures and equations in an order which distract from the core message of the paper.*

*1. The goal of the paper appears to be to establish a methodology for using interferometric analysis of repeat SAR data to estimate high resolution aggradation/subsidence rates in an alluvial environment. However, the results presented are not compared with any alternative estimates of topographic change in the study area, and so the validity of the results are unclear. A comparison of the estimated aggradation rates for other dissimilar fluvial environments like the lower Ganga River and the Floodplain of the upper Yamuna Valley are presented as being on the same order of magnitude as their results, but this does little to support the methodology. This technique might provide an amazing new tool for assessing high-resolution topographic changes associated with fluvial processes using publicly available data (awesome!), but a lack of ground truthing or valid comparison with other methods for the results really hurts the papers ultimate impact. If no further validation is possible, I think the paper needs to highlight the preliminary nature of the results so as*

*not to be cited as a source for known aggradation/degradation rates, and soften statements such as "We successfully mapped millimeter-scale elevation changes in (four) river channel(s) over a ~15km stretch…"*

AR: Since this is the first observation detecting millimeter-scale riverbed elevation changes, there are no existing methods to compare with. One approach is to use the subsidence rate in the adjacent cropland, mapped using the conventional InSAR method based on the deformation phase. Whether the conventional InSAR deformation phase or the residual topographic phase from our new approach is used, as long as the input phase data comprises high quality, the results should indicate the same rate. For instance, in Figure 17, point C, which is adjacent to river 3, the data shows a subsidence rate of -14 mm/year based on conventional InSAR deformation phase. In Figure 14, river 3 near point C, the data indicate a subsidence rate of -12 mm/year, with an uncertainty range of -12% to +8%, corresponding to rates between -13.4 mm/year and -11 mm/year based on the differential residual topographic phase.

This novel approach works with the residual topographic phase, making the quality of the topographic phase a fundamental factor. The topographic phase is caused by variations in terrain; hence, flat terrain does not exhibit a topographic phase. For mountainous terrain, the applicability needs further testing to determine the limits of slope regularity and angle. In this initial study, the four rivers exhibit consistent slopes, with a slope gradient of 0.008 (vertical: horizontal). This low-angle slope results in a low phase gradient, which contributes to a good quality of residual topographic phase, with no challenge in its unwrapping as well. To ensure the quality of the residual topographic phase, the phase profile along the river slope is an important metric to assess. If an area is relatively flat and small, it may not provide a clear topographic phase trend profile for quality control, making this novel approach potentially inapplicable in such cases.

From Figure 4(b), the coherence at a pixel located at the mountain front of river 2 shows distinct seasonal patterns. The highest coherence occurs around January each year, indicating the driest period with minimal disturbance on the riverbeds' surface. Conversely, the lowest coherence occurs around June, when the bumpy river water surface causes reduced coherence. Although there are slight variations between years from 2017 to 2021, the overall seasonal sinusoidal pattern and amplitude remain consistent. This highlights an important point: the coherence time-series reflects the regular cycle of river inundation and drying, which may suggest consistent sediment aggradation rates from these natural cycles. However, if this cycle is disrupted by an extreme flood event, the annual sediment aggradation rates would likely be affected by the abrupt sedimentation from such an extreme event.

*2. The paper is challenging to read due to repetitious information and confusing narrative structure, grammatical issues, calls to figures that don't exist (e.g. figure 18), and calling the figures and equations in a loosely structured order.*

AR: Thanks, now the numbers are corrected and we have worked through the manuscript to check for grammar and the narrative. We believe condensing the methodological context at the beginning helps.

*Figures: To be clear, I don't believe all figures and equations must only ever be called in order, but the paper would benefit greatly from trying to streamline the readers experience somewhat. The current structure results in a lot of searching around trying to match up the text with the*

*figures/equations. Additionally, many figures include imbedded text for the axes and legends that are so small they are either challenging to read when printed, or are just unreadable even when zooming in on a computer.*

AR: Figures 8 and 9 have been updated to remove the small embedded text. Their captions have also been revised to include explanations of what the small embedded text represents within the figures.

*Grammar and consistency: There are many examples of bad grammar and sloppy editing which make the paper challenging to read. For example: using the acronym "LOS" on line 241 and in Figure 9, but defining it on line 328, in Figure 12, then again on line 383, and 467. An example of a grammatical issue is: "Azimuth is along satellite fly track direction, range is across satellite fly track direction". The paper would greatly benefit from a copy edit. I have included many examples of copy editing issues in my line edits, but they are concentrated at the start of the paper, and are not comprehensive.*

AR: Now 'LOS' is defined on line 244.

AR:
| | |
|---|---|
| | the DEM is outdated, it will include residual topographic phase mixed with the line-of-sight (LOS) motion phase as input for |
| 245 | SBAS inversion, leading to higher uncertainty in the LOS displacement results (Berardino et al., 2002; Morishita et al., 2020). |

*Narrative structure: The methodological context followed by the authors specific methods results in some repetition, and there are concepts in the methodological context which do not seem to impact the core thrust of the paper. It seems to me that it could be shortened and focused, although this may just be a product of my lack of experience in the field of satellite based remote sensing.*

AR: We have now explained in lines 161–166 why certain concepts in the methodological context impact the core of the paper. We have also put some of this methodological context into the Supplementary Material 1.

AR:
| | |
|---|---|
| 160 | ripples. Since the amplitude is average in dry season of year 2019, the effect of soil moisture of the sandy riverbed might be less significant. Because the SBAS-InSAR method relies on distributed backscatter, which is particularly effective in areas with diffuse scattering with stronger VV polarization, it is important to explain upfront the backscattering type and polarization characteristics of the dry gravel riverbeds. Additionally, applying this novel approach to more complex rivers, beyond ephemeral rivers, requires classifying dry gravel pixels based on SAR amplitude polarization characteristics and statistical |
| 165 | metrics. It is important to use SAR amplitude for classification instead of optical or multi-spectral images, as the same SAR images' phase component is used in this novel approach to map sediment aggradation rates. |

*3. My understanding after reading the paper is that you do interferograms during the dry season, and across the monsoon season, but are unable to resolve interferograms during the monsoon season. However, I'm still not entirely sure how the authors deal with phase ambiguity if changes in elevation that occur across the monsoon season are quite large. Is my understanding correct? Is there data loss in the interannual interferograms associated with phase ambiguity? Can you provide a length scale of observable movement before the phase ambiguity is surpassed? Additionally, the authors see a fair amount of aggradation during the dry seasons in the river channel (fig 16), and explain this as "due to a combination of noise and varying perpendicular baselines caused topographic sensitivity variation." I don't understand that explanation, especially when the magnitude of the aggradation is so consistent, doesn't look like a noise signal, and sometimes looks*

*like it has a similar slope to your inter-annual aggradation rates. Can you clarify my misunderstanding of the work, or address this concern with the data?*

AR: The relatively large gap during the monsoon season is not problematic because the phase information used to fill this gap combines phase ambiguity and sediment aggradation caused phase difference. The phase ambiguity is not removed and is represented as small jumps, as shown in Figure 16. These small jumps contribute to the uncertainty range of -12% to +8%.

For example, in the black points from the 2017 time-series in Figure 16, the ±100 m perpendicular baseline introduces a phase ambiguity of 0–5 mm/yr. Therefore, observable movement of 16 mm/yr cross the larger interferogram network gap, is required to exceed this 5 mm/yr phase ambiguity. In general, points of different colours within the same year should exhibit similar phase ambiguity caused small jumps because they share the same perpendicular baseline network, as illustrated in Figure 8. However, some variations in small jumps between different coloured points within the same year are observed. These discrepancies may be attributed to atmospheric noise and/or other factors. This might explain the irregular jumps between flow distances of 142,000–134,000 m in Figure 14. In this study, only the gaps between the two point clouds are considered as signal. The other possible factors and its mechanisms are worth to look at in the future studies.

Future studies will address the phase ambiguity issue. This study focuses on demonstrating a novel approach to using differential residual topographic phase to map river sediment aggradation rates. In our opinion, it is necessary to understand the original residual topographic phase history before applying further processes, such as flattening small jumps. Advancing this approach will be the immediate focus of future research. Figure 10(b) potentially illustrates the phase difference between two different years (black and blue), providing a more straightforward representation of phase differences caused by river aggradation.

*Line edits:*

*L36: "where river channels"*

*L66: "and are labelled rivers 1-4"*

*Figure 1: "are approximately 15 km in length, and 300 m in width"*

*L107: "bars result in stronger"*

*L117: "Azimuth is along the satellite flight path, and range is perpendicular to the satellite flight path"*

*L121: "Amplitude is calculated following Eq. (5-8)"*

AR: Updated now, thanks!

*Figure 2: Why are the bar graphs in e-f descending rather than ascending. Not sure if this is some standard I'm unaware of, but it confused me.*

AR: The SAR amplitude values in decibels typically range from about -25 dB to 0 dB for the most cases. This is why the bar graphs in e-f descending. We have now updated the figure caption to point out this information.

AR:
125

**Figure 2**: Illustration of the SAR diffuse backscatter from a dry gravel riverbeds. **The returned SAR backscatter energy intensity is usually very small because the radar signal loses strength as it travels to the target and back. The decibel (dB) scale is logarithmic, and the logarithm of a small number (less than 1) is negative. The SAR amplitude values in decibels typically range from about -25 dB to 0 dB for most cases (Flores-Anderson et al., 2019).** This illustration demonstrates that each 20 m² InSAR mapping pixel contains tens of thousands of distributed scatters. The intensity of the SAR signal represents the sum of the distributed scatters

*Figure 3: I recommend bigger text, hard to read.*

*Figure 4: bigger text*

*L241: Define "LOS"*

*Figure 5,6,7: bigger text*

*Figure 8,9: way bigger text for the legend!*

*L330: Figure 18?*

*Figure 10, 11, 12, 13: Bigger text*

AR: Updated now, thanks!

*Figure 14: it's interesting that you have larger uncertainty for river 1 and 2 which have more consistent data, and less uncertainty in river 3 and 4 which are much more noisy looking. Can you comment on this? Text size is ok here, but could still be bigger.*

AR: In Figure 14, the shaded in grey spans from -12% to +8% is for every dots in the plot. Somehow when the dots are less scattered (e.g. river 1 and 2), the plot of shaded in grey shows the range really well. When the dots are scattered (e.g. river 3 and 4), the plot of shaded in grey not display well with the shaded range. However, the uncertainty range of -12% to +8% is the same at every dots in the elevation change rates plot in Figure 14.

*L445: Figure 18!*

*Figure 16, 17: Bigger text. Also where is location E on the map?*

*L468: Figure 18!*

AR: Updated now, thanks!

*Section 8.3: This is a very interesting hypothesis about the relation between aggradation rates and avulsion timing. Can you find any evidence to support that in the sediment record near these rivers, or just another citation that might support this idea?*

AR: We agree that this is a very interesting avenue for continued research. We have added three additional references that discuss rapid avulsion frequencies in rivers immediately east (the Kosi) and west (the Bagmati) of our study site in lines 499-503. The rates they quote are faster than the estimates we gave and that may be more closely linked to the behaviour at avulsion nodes rather than just a function of elevation contrast.

AR:

500    avulsion every few hundreds of years (i.e. channel depth divided by aggradation rate). However, other mechanisms such as a sudden reduction in transport capacity near the avulsion node may cause the river to spill and avulse (Jones and Schumm, 1999). The Bagmati River which is just west of our study site in the Gangetic Plains has been described as 'hyper-avulsive' and has a record of channel avulsion on a decadal to century scale (Jain and Sinha, 2003; Sinha et al., 2005). Similar avulsion frequencies have also been recorded over the large Kosi River that drains east of our study area (Chakraborty et al., 2010).